# The Pitfalls of Memorization: When Memorization Hurts Generalization

**Reza Bayat**[*1,2], **Mohammad Pezeshki**[*3]
**Elvis Dohmatob**[1,3,4] , **David Lopez-Paz**[3] , **Pascal Vincent**[1,2,3]
[1]Mila    [2]Université de Montréal    [3]Meta FAIR    [4]Concordia University

## Abstract

Neural networks often learn simple explanations that fit the majority of the data while memorizing exceptions that deviate from these explanations. This behavior leads to poor generalization when the learned explanations rely on spurious correlations. In this work, we formalize *the interplay between memorization and generalization*, showing that spurious correlations would particularly lead to poor generalization when are combined with memorization. Memorization can reduce training loss to zero, leaving no incentive to learn robust, generalizable patterns. To address this, we propose *memorization-aware training* (MAT), which uses held-out predictions as a signal of memorization to shift a model's logits. MAT encourages learning robust patterns invariant across distributions, improving generalization under distribution shifts.

## 1 Introduction

Watching the stars at night gives the illusion that they orbit around the Earth, leading to the longstanding belief that our planet is the center of the universe—a view supported by great philosophers such as Aristotle and Plato. This Earth-centric model could explain the trajectories of nearly all celestial bodies. However, there were exceptions; notably, five celestial bodies, including Mars and Venus, occasionally appeared to move backward in their trajectories (Kuhn, 1992). To account for these anomalies, Egyptian astronomer Claudius Ptolemy introduced "epicycles"—orbits within orbits—a complex yet effective system for predicting these movements. Over 1400 years later, Nicolaus Copernicus proposed an alternative model that placed the Sun at the center. This Sun-centric view not only simplified the model but also naturally explained the previously perplexing backward motions.

Drawing a parallel to modern machine learning, we observe a similar phenomenon in neural networks. Just as the Earth-centric model provided a incomplete explanation that required epicycles to account for exceptions, neural networks can learn simple explanations that work for the majority of their training data (Geirhos et al., 2020; Shah et al., 2020; Dherin et al., 2022). These models might then treat minority examples—those that do not conform to the learned explanation—as exceptions (Zhang et al., 2021). This becomes particularly problematic if the learned explanation is spurious, meaning it does not hold in general or is not representative of the true data distribution (Idrissi et al., 2022; Sagawa et al., 2020; Pezeshki et al., 2021; Puli et al., 2023).

Empirical Risk Minimization (ERM), the standard learning algorithm for neural networks, can exacerbate this issue. ERM enables neural networks to quickly capture spurious correlations and, with sufficient capacity, memorize the remaining examples rather than learning the true patterns that explain the entire dataset. This has real-world implications; for example, neural networks designed to detect COVID-19 from x-ray images have been found to rely on spurious correlations, such as whether a patient is standing or lying down (Roberts et al., 2021). This could be dangerously misleading, as a model that appears to excel in most cases may have actually captured a spurious correlation. Memorization of the remaining minority examples can **fully mask a neural network's failure** to grasp the true patterns in the data, giving a false sense of reliability. Again, the Earth-centric model

---

[*]Equal contribution.
Code: https://github.com/facebookresearch/Pitfalls-of-Memorization

of the universe was able to explain all celestial trajectories with complex epicycles but ultimately failed to reveal the true nature of our solar system.

Identifying whether a model with nearly perfect accuracy on the training data has learned generalizable patterns or merely relies on a mix of spurious correlations and memorization is critical. The answer lies in the model's performance on **held-out** data, particularly on minority examples. Metrics such as held-out average accuracy or more fine-grained group accuracies can help us identify a better model. A question that arises is: *How can one use held-out performance signals to proactively guide a model toward learning generalizable patterns?*

Traditionally, held-out performance signals are mainly used for hyperparameter tuning and model selection. However, in this work, we propose a novel approach that leverages these signals strategically to guide the learning process. Towards this goal, our paper makes the following contributions:

- *Formalizing the interplay between memorization and generalization*: We study how memorization affects generalization in an interpretable setup. We show that while spurious correlations are inherently problematic, they *by themselves* do not always lead to poor generalization in neural networks. Instead, it is the combination of spurious correlations with memorization that leads to this problem. Our analysis shows that models trained with empirical risk minimization (ERM) tend to rely on spurious features for the majority of the data while memorizing exceptions, achieving zero training loss but failing to generalize on minority examples.

- *Introducing memorization-aware training (MAT)*: MAT is a novel learning algorithm that leverages the flip side of memorization by using held-out predictions to shift a model's logits during training. This shift guides the model toward learning invariant features that generalize better under distribution shifts. Unlike ERM, which relies on the i.i.d. assumption, MAT is built upon an alternative assumption that takes into account the instability of spurious correlations across different data distributions.

## 2 THE INTERPLAY BETWEEN MEMORIZATION AND SPURIOUS CORRELATIONS IN ERM

**Problem Setup and Preliminaries.** We consider a standard supervised learning setup for a $K$-class classification problem. The data consists of input-label pairs $\{(\boldsymbol{x}_i, y_i)\}_{i=1}^n$, where $\boldsymbol{x}_i$ is the input vector and $y_i \in \{1, \ldots, K\}$ is the class label. Let $a_i$ denote any attribute or combination of attributes within $\boldsymbol{x}_i$ that may or may not be relevant for predicting the target $y_i$. The objective is to train a model $\hat{p}(y \mid \boldsymbol{x}; \boldsymbol{w})$ parameterized by $\boldsymbol{w}$. Given an input $\boldsymbol{x}_i$, let $f(\boldsymbol{x}_i; \boldsymbol{w}) \in \mathbb{R}^K$ represent the output logits of the model, then:

$$\hat{p}(y \mid \boldsymbol{x}_i; \boldsymbol{w}) = \text{softmax}(f(\boldsymbol{x}_i; \boldsymbol{w})). \tag{1}$$

Under the i.i.d. assumption that $p(y, \boldsymbol{x})$ is invariant between training and test sets, empirical risk minimization (ERM) seeks to minimize the following loss over the training dataset:

$$\mathcal{L}^{\text{ERM}} = \frac{1}{n} \sum_{i=1}^n l(\hat{p}(y \mid \boldsymbol{x}_i; \boldsymbol{w}), y_i) + \frac{\lambda}{2} ||\boldsymbol{w}||^2, \tag{2}$$

where $l(., .)$ is the cross-entropy loss, and $\frac{\lambda}{2}||\boldsymbol{w}||^2$ is the weight-decay regularization.

### 2.1 MEMORIZATION CAN EXACERBATE SPURIOUS CORRELATIONS

Spurious correlations violate the i.i.d. assumption, and when combined with memorization, can hurt generalization. We now study such scenario. Adapting the frameworks introduced in Sagawa et al. (2020) and Puli et al. (2023), we look into the interplay between memorization and spurious correlations in an interpretable setup.

**Setup 2.1** (Spurious correlations and memorization). *Consider a binary classification problem with labels $y \in \{-1, +1\}$ and an unknown spurious attribute $a \in \{-1, +1\}$. Each input $\boldsymbol{x} \in \mathbb{R}^{d+2}$ is given by $\boldsymbol{x} = (x_y, \gamma x_a, \boldsymbol{\epsilon})$, where $x_y \in \mathbb{R}$ is a core feature dependent only on $y$. $x_a \in \mathbb{R}$ is a spurious feature dependent only on $a$, and $\boldsymbol{\epsilon} \in \mathbb{R}^d$ are example-specific noise features uncorrelated with both $y$*

and $a$. The scalar $\gamma \in \mathbb{R}$ modulates the rate at which the model learns to rely on the spurious feature $x_a$, effectively acting as a scaling factor that increases the feature's learning rate relative to the core feature $x_y$. The attribute $a$ is considered spurious; it is assumed to be correlated with the labels $y$ at training but has no correlation with $y$ at test time, potentially leading to poor generalization if the model relies on $x_a$. Specifically, the data generation process is defined as:

$$
\boldsymbol{x} = \begin{bmatrix} x_y \sim \mathcal{N}(y, \sigma_y^2) \\ \gamma x_a \sim \mathcal{N}(a, \sigma_a^2) \\ \boldsymbol{\epsilon} \sim \mathcal{N}(0, \sigma_{\boldsymbol{\epsilon}}^2 \boldsymbol{I}) \end{bmatrix} \in \mathbb{R}^{d+2} \ \ where, \ a = \begin{cases} y & w.p. \ \rho \\ -y & w.p. \ 1-\rho \end{cases} and \ \rho = \begin{cases} \rho^{tr} & \textit{(train)} \\ 0.5 & \textit{(test)} \end{cases}. \quad (3)
$$

To better understand this setup, one can think of a classification task between cows and camels. In this example, $\boldsymbol{x}$ represents the pixel data, $y \in \{$cow, camel$\}$ are the class labels, and $a \in \{$grass, sand$\}$ are the background labels. Here, $x_y$ represents the pixels associated with the animal itself (either cow or camel), $x_a$ represents the pixels associated with the background (grass or sand), and $\boldsymbol{\epsilon}$ represents irrelevant pixels that varies from one example to another. The **key assumption** is that the joint distribution of class labels and attribute labels differs between training and test datasets, i.e., $p^{tr}(a, y) \neq p^{te}(a, y)$. For example, in the training set, most cows (camels) might appear on grass (sand), while in the test set, cows (camels) appear equally on each background.

**Illustrative Scenarios.** We first empirically study a configuration of the above setup where $\rho^{tr} = 0.9$ makes the $a$ spuriously correlated with $y$. We set $\gamma = 5$ making the spurious feature easier for the model to learn. In contrast, the core feature $x_y$ is fully correlated with $y$, but due to a smaller norm, it is learned more slowly. Here we consider two cases:

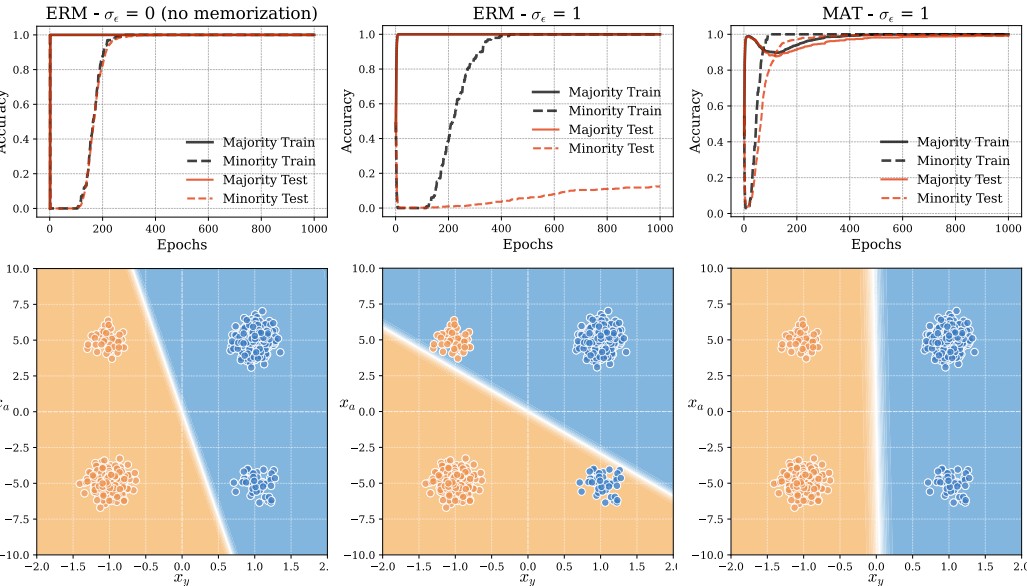

Figure 1: Illustration of two scenarios in the interpretable classification setup involving spurious correlations and memorization. The left panel represents a scenario without example-specific features ($\sigma_{\boldsymbol{\epsilon}} \to 0$), where memorization is not possible. In this case, the model trained with ERM initially learns the spurious feature $x_a$ serving the majority, but eventually adjusts the decision boundary to the core feature $x_y$, resulting in good generalization on minority test examples. The middle and right panels depict a scenario with example-specific features ($\sigma_{\boldsymbol{\epsilon}} > 0$), where memorization is possible. In the middle plot, the model trained with ERM fails to generalize as it memorizes the minorities using the example-specific features $\boldsymbol{\epsilon}$ leaving no more incentive for the model to learn the core feature. In contrast, the model trained with MAT successfully learns the invariant features, and generalizes well even in the presence of example-specific features.

1. ***No example-specific features $\Rightarrow$ No Memorization $\Rightarrow$ ERM generalizes well.*** Figure 1-(left) presents a case where there are no input example-specific features ($\sigma_\epsilon \to 0$). As training progresses, the model first learns $x_a$ due to its larger norm, resulting in perfect accuracy on the majority examples. Once the model achieve nearly perfect accuracy on the majority examples, it starts to learn the minority examples. At this point, the model must adjust its decision boundary to place more emphasis on the core feature $x_y$, ultimately achieving perfect generalization on both majority and minority examples.

2. ***With example-specific features $\Rightarrow$ Spurious Features + Memorization $\Rightarrow$ ERM fails to generalize.*** Figure 1-(middle) presents a similar setup to the former, but this time with input example-specific features ($\sigma_\epsilon > 0$). Again, initially, the model learns to rely on the spurious feature $x_a$. However, unlike Case 1, the example-specific features $\epsilon$ provides the model an opportunity to memorize minority examples directly. As a result, the model achieves zero training loss by memorizing minority examples using the example-specific dimensions instead of learning to rely on the core feature $x_y$. Consequently, the model fails to adjust its decision boundary to align with $x_y$, and does not generalize on held-out minority examples.

These results illustrate that the combination of spurious correlations and memorization creates a 'loophole' for the model. When memorization happens, it leaves **no more incentives** for the model to learn the true, underlying patterns necessary for robust generalization.

**Theoretical Analysis.** We now provide a formal analysis to formalize our empirical observations. Complete proofs are provided in Appendix F.

**Theorem 2.2.** *Consider a binary classification problem under the setup described in Setup 2.1, where a linear model $f(\boldsymbol{x}; \boldsymbol{w}) = \boldsymbol{x}^\top \boldsymbol{w}$ is trained using ERM on a dataset $\mathcal{D}^{tr}$. Let $\widehat{\boldsymbol{w}}_{ERM} = (\widehat{w}_y, \widehat{w}_a, \widehat{\boldsymbol{w}}_\epsilon) \in \mathbb{R}^{d+2}$ be the learned parameters, and $\widehat{y}(\boldsymbol{x}) = \operatorname{argmax}_y f_y(\boldsymbol{x}; \widehat{\boldsymbol{w}}_{ERM})$ the learned classifier.*

*Under the conditions where, $\lambda \to 0^+$, $n \to \infty$, $\lambda\sqrt{n} \to \infty$, $\rho^{tr} > 0.501$, we have,*

0. ***Perfect training accuracy:*** *The classifier achieves perfect accuracy on all training examples:*
$$p\left(\widehat{y}(\boldsymbol{x}) = y\right) \to 1, \quad \forall \boldsymbol{x} \in \mathcal{D}^{tr}.$$

1. ***No example-specific features:*** *If $\sigma_\epsilon \to 0^+$, the classifier converges to one that relies solely on the core feature $x_y$. For a test point $\boldsymbol{x}$:*
$$p\left(\widehat{y}(\boldsymbol{x}) = y\right) \to 1, \quad \forall \boldsymbol{x} \notin \mathcal{D}^{tr}.$$

2. ***With example-specific features:*** *If $\sigma_\epsilon > 0$ is bounded away from zero and $d \gg \log n$ and $\gamma \gg \sigma_\epsilon\sqrt{d/m}$, where $m := \rho^{tr} n$ is the number of majority samples in the training set. Then, for a test point $\boldsymbol{x}$, the classifier relies pathologically on the spurious feature $x_a$, i.e.,*
$$p\left(\widehat{y}(\boldsymbol{x}) = a\right) \to 1, \quad \forall \boldsymbol{x} \notin \mathcal{D}^{tr}.$$

*The condition $d \gg \log n$ ensures that example-specific features from different samples are approximately orthogonal, and $\gamma \gg \sigma_\epsilon\sqrt{d/m}$ guarantees that the spurious feature $x_a$ is learned faster by gradient descent than other features.*

Theorem 2.2 suggests that in the presence of spurious correlations and example-specific features, $p(\widehat{y}(\boldsymbol{x}) = a) \to 1$ for held-out examples. This indicates that the model confidently predicts $a$ as the label for any held-out input $\boldsymbol{x} \notin \mathcal{D}^{tr}$, leading to poor generalization. The probability that a model assigns to the held-out examples is then defined as:

$$p(y^{ho} \mid \boldsymbol{x}) = \operatorname{softmax}\left(\frac{f(\boldsymbol{x}; \widehat{\boldsymbol{w}}_{ERM})}{\tau}\right) \quad \forall \boldsymbol{x} \notin \mathcal{D}^{tr}, \tag{4}$$

where the superscript 'ho' stands for held-out and $\tau > 0$ is a temperature parameter that controls the sharpness of the distribution. This probability serves as the basis for the shifts introduced in the next section.

## 3 MEMORIZATION-AWARE TRAINING (MAT)

As exemplified in the previous section, spurious correlations between labels $y$ and certain attributes $a$ can lead to poor generalization, particularly when models memorize specific training examples rather than learning robust patterns. To mitigate this issue, we propose *memorization-aware training (MAT)*, which leverages calibrated held-out probabilities to shift the logits, **prioritizing learning of examples with worse generalization performance**.

### 3.1 SHIFTING THE LOGITS WITH CALIBRATED HELD-OUT PROBABILITIES

MAT modifies the empirical risk minimization (ERM) objective by introducing a per-example logit shift based on calibrated probabilities. The training loss is redefined as:

$$\mathcal{L}^{\mathrm{MAT}} = \frac{1}{n} \sum_{i=1}^{n} l\big(\mathrm{softmax}(f(\boldsymbol{x}_i; \boldsymbol{w}) + \log \overline{p}^{ho}(\cdot \mid \boldsymbol{x}_i)), y_i\big), \tag{5}$$

where $f(\boldsymbol{x}_i; \boldsymbol{w})$ are the logits for input $\boldsymbol{x}_i$, and $\overline{p}^{ho}(\cdot \mid \boldsymbol{x}_i)$ are the calibrated held-out probabilities, defined as:

$$\overline{p}^{ho}(y \mid \boldsymbol{x}) := \sum_{y^{ho}} p(y \mid y^{ho}) p(y^{ho} \mid \boldsymbol{x}), \tag{6}$$

where $p(y^{ho} \mid \boldsymbol{x})$ is derived from Eq. equation 4, and $p(y \mid y^{ho})$ is the calibration matrix, given by:

$$p(y \mid y^{ho}) = \frac{p(y, y^{ho})}{\sum_{y'} p(y', y^{ho})}, \quad \text{where} \quad p(y, y^{ho}) = \frac{1}{n} \sum_{\{i : y_i = y\}} p(y^{ho} \mid \boldsymbol{x}_i). \tag{7}$$

The calibration matrix $p(y \mid y^{ho})$ represents the confusion matrix of true labels $y$ versus held-out probabilities $y^{ho}$. The calibrated probabilities $p(y^{ho} \mid \boldsymbol{x})$ guide the learning such that correct held-out predictions lower the loss for an example, while incorrect predictions increase it, prompting the model to focus on challenging examples. This ensures MAT prioritizes minority examples with poor generalization, improving generalization under distribution shifts. See Appendix B for a discussion on the effect of loss reweighting and logit shifting.

**Estimating** $p(y^{ho} \mid \boldsymbol{x}_i)$**.** The final challenge in implementing MAT is that we require an auxiliary model to provide held-out probabilities for every single example. This auxiliary model can be derived using Cross-Risk Minimization (XRM) (Pezeshki et al., 2023).

XRM, originally proposed as an environment discovery method, trains two networks on random halves of the training data, encouraging each to learn a biased classifier. It uses cross-mistakes (errors made by one model on the other's data) to annotate training and validation examples. Here, MAT does not require environment annotations but uses held-out logits, $f^{ho}(\boldsymbol{x})$, from a pretrained XRM model. For model selection using the validation set, either ground-truth or inferred environment annotations from XRM can be used.

The pseudo-code in Algorithm 1 summarizes the MAT algorithm. Note that MAT introduces a single hyper-parameter $\tau$, the softmax temperature in Eq. equation 4.

## 4 EXPERIMENTS

We validate the effectiveness of MAT in improving generalization under subpopulation shift. Then, we provide a detailed analysis of the memorization behaviors of models trained with ERM and MAT.

### 4.1 EXPERIMENTS ON SUBPOPULATION SHIFT

We evaluate our approach on four datasets under subpopulation shift, as detailed in Appendix A. In all experiments, we assume that training environment annotations are not available. For the validation set and for the purpose of model selection, we consider two settings: (1) group annotations are available in the validation set for model selection, and (2) no annotations are available even in the validation set.

---

**Algorithm 1** Memorization-Aware Training (MAT)

---

**Input:** Training set $\{(\boldsymbol{x}_i, y_i)\}_{i=1}^n$, Validation set $\{(\boldsymbol{x}_i', y_i')\}_{i=1}^m$, Pre-trained XRM model $f^{\mathrm{XRM}}(\boldsymbol{x})$

**Optional:** Validation environment annotations $\{a_i'\}_{i=1}^m$; if not available, infer $\tilde{a}_i' = \operatorname{argmax} f^{\mathrm{ho}}(\boldsymbol{x}_i) \neq y_i$

**Model Selection:** Early stopping based on validation worst-group accuracy (using $a_i'$ if provided, or $\tilde{a}_i'$ if inferred)

- Initialize a classifier $f(\boldsymbol{x})$.
- Compute $p(y^{\mathrm{ho}} \mid \boldsymbol{x}_i) = \operatorname{softmax}(f^{\mathrm{XRM}}(\boldsymbol{x}_i)/\tau)$.
- Compute $p(y, y^{\mathrm{ho}}) = \operatorname{concat}\left(\frac{1}{n} \sum p(y^{\mathrm{ho}} \mid \boldsymbol{x}_i)[y = y_i]\right)_{y_i \in \{1,\ldots,K\}}$
- Compute $p(y \mid y^{\mathrm{ho}}) = \frac{p(y, y^{\mathrm{ho}})}{\langle p(y, y^{\mathrm{ho}}), \mathbf{1} \rangle}$
- Compute $\overline{p}^{\mathrm{ho}}(y \mid \boldsymbol{x}_i) = \langle p(y^{\mathrm{ho}} \mid \boldsymbol{x}_i), p(y \mid y^{\mathrm{ho}})^T \rangle$
- Repeat until early stopping:
  - Update the loss: $\frac{1}{n} \sum l(\operatorname{softmax}(f(\boldsymbol{x}_i) + \log \overline{p}^{\mathrm{ho}}(. \mid \boldsymbol{x}_i), y_i)$
  - Track worst-group accuracy and update the best model
  - Stop if no improvement is observed after $P$ iterations

---

Table 1: Average and worst-group accuracies (avg/wga) comparing methods. We specify access to group annotations in training $(tr)$ and validation $(va)$ data. Symbol † denotes original numbers.

| $tr$ | $va$ | | Waterbirds | | CelebA | | MultiNLI | | CivilComments | |
|---|---|---|---|---|---|---|---|---|---|---|
| | | | Avg | WGA | Avg | WGA | Avg | WGA | Avg | WGA |
| ✓ | ✓ | GroupDRO | $90.2_{\pm 0.3}$ | $86.5_{\pm 0.5}$ | $93.1_{\pm 0.3}$ | $88.3_{\pm 2.1}$ | $80.6_{\pm 0.4}$ | $73.4_{\pm 4.8}$ | $84.2_{\pm 0.2}$ | $73.8_{\pm 0.6}$ |
| ✗ | ✓ | ERM | 97.3 | 72.6 | 95.6 | 47.2 | 82.4 | 67.9 | 83.1 | 69.5 |
| | | LFF† | 91.2 | 78.0 | 85.1 | 77.2 | 80.8 | 70.2 | 68.2 | 50.3 |
| | | JTT† | 93.3 | 86.7 | 88.0 | 81.1 | 78.6 | 72.6 | 83.3 | 64.3 |
| | | LC† | - | $90.5_{\pm 1.1}$ | - | $88.1_{\pm 0.8}$ | - | - | - | $70.3_{\pm 1.2}$ |
| | | AFR† | $94.2_{\pm 1.2}$ | $90.4_{\pm 1.1}$ | $91.3_{\pm 0.3}$ | $82.0_{\pm 0.5}$ | $81.4_{\pm 0.2}$ | $73.4_{\pm 0.6}$ | $89.8_{\pm 0.6}$ | $68.7_{\pm 0.6}$ |
| | | MAT | $90.4_{\pm 0.7}$ | $88.1_{\pm 0.9}$ | $92.4_{\pm 0.4}$ | $90.5_{\pm 1.0}$ | $79.4_{\pm 0.4}$ | $74.6_{\pm 1.0}$ | $84.3_{\pm 0.3}$ | $74.0_{\pm 0.8}$ |
| ✗ | ✗ | ERM | 83.5 | 66.4 | 95.4 | 54.3 | 82.1 | 67.9 | 81.3 | 67.2 |
| | | uLA† | $91.5_{\pm 0.7}$ | $86.1_{\pm 1.5}$ | $93.9_{\pm 0.2}$ | $86.5_{\pm 3.7}$ | - | - | - | - |
| | | XRM† | $89.3_{\pm 0.6}$ | $88.1_{\pm 0.9}$ | $91.4_{\pm 0.5}$ | $89.1_{\pm 1.3}$ | $75.8_{\pm 1.2}$ | $72.1_{\pm 1.0}$ | $84.4_{\pm 0.6}$ | $72.2_{\pm 0.8}$ |
| | | MAT | $90.4_{\pm 0.7}$ | $88.1_{\pm 0.9}$ | $92.3_{\pm 0.3}$ | $89.9_{\pm 1.2}$ | $79.6_{\pm 0.2}$ | $73.0_{\pm 0.8}$ | $85.7_{\pm 0.1}$ | $68.1_{\pm 0.7}$ |

For evaluation, we report two key metrics on the test set: (1) average test accuracy and (2) worst-group test accuracy, the latter being computed using ground-truth annotations.

Table 1 compares the performance of MAT with several baseline methods, including ERM, GroupDRO (Sagawa et al., 2019), and other invariant methods like LfF (Nam et al., 2020), JTT (Liu et al., 2021), LC (Liu et al., 2022), uLA (Tsirigotis et al., 2024), AFR (Qiu et al., 2023), XRM+GroupDRO (Pezeshki et al., 2023). These methods vary in their assumptions about access to annotations, both in training and validation for model selection. For instance, ERM does not assume any training group annotations, while GroupDRO has full access to group annotations for training and validation data. Further details on the experimental setup and methods are in Appendix A.

## 4.2 ANALYSIS OF MEMORIZATION SCORES

To understand the extent of memorization in models trained with ERM, we analyze the distribution of memorization scores across subpopulations. We focus on the Waterbirds dataset, which includes two main classes—Waterbird and Landbird —each divided into majority and minority subpopulations based on their background (e.g., Waterbird on water vs. Waterbird on land).

The memorization score is derived from the influence function, which measures the effect of each training sample on a model's prediction. Formally, the influence of a training sample $i$ on a target

sample $j$ under a training algorithm $\mathcal{A}$ is defined as:

$$\text{infl}(\mathcal{A}, \mathcal{D}, i, j) := \hat{p}_{\mathcal{D}}^{(\mathcal{A})}(y_j \mid \boldsymbol{x}_j) - \hat{p}_{\mathcal{D}_{\neg(\boldsymbol{x}_i, y_i)}}^{(\mathcal{A})}(y_j \mid \boldsymbol{x}_j) \tag{8}$$

where $\mathcal{D}$ is the training dataset, $\mathcal{D}_{\neg(\boldsymbol{x}_i, y_i)}$ denotes the dataset with the sample $(\boldsymbol{x}_i, y_i)$ removed. The memorization score is a specific case of this function where the target sample $(\boldsymbol{x}_j, y_j)$ is the same as the training sample. It measures the difference between a model's performance on a training sample when that sample is included in the training set (held-in) versus when it is excluded (held-out).

Calculating self-influence scores using a naive leave-one-out approach is computationally expensive. However, recent methods, such as TRAK (Park et al., 2023), provide an efficient alternative. TRAK approximates the data attribution matrix, and the diagonal of this matrix directly gives the self-influence scores (see Appendix A.2 for more details).

Figure 2 depicts the distribution of self-influence scores across subpopulations in the Waterbird dataset. We note that minority subpopulations (e.g., Waterbirds on land) show higher self-influence scores compared to their majority counterparts (e.g., Waterbirds on water) in a model trained with ERM. However, a model trained with MAT shows a similar distribution of self-influence scores for both the majority and minority examples, with overall lower scores compared to ERM. These results show that MAT effectively reduced memorization, while leading to improved generalization.

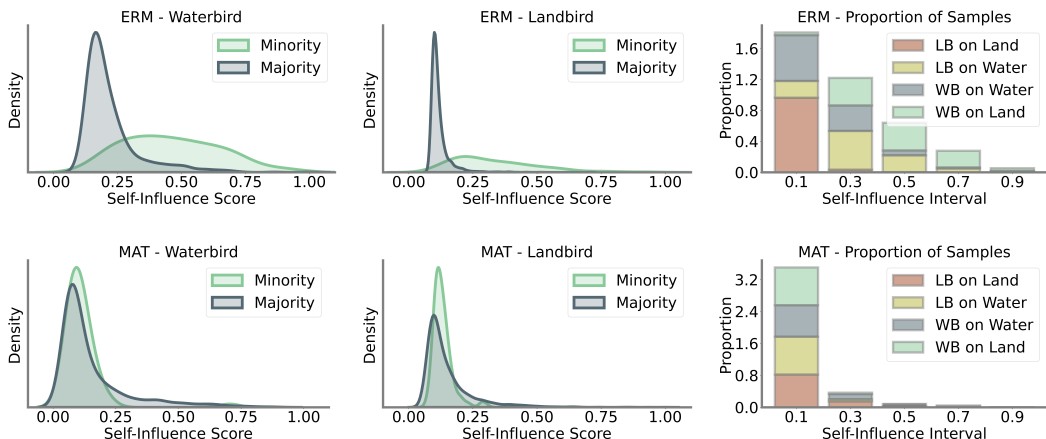

Figure 2: Self-Influence estimation of the Waterbird groups by ERM and MAT. The distribution of self-influence scores is shown for both the majority and minority subpopulations (e.g., Waterbirds on water vs. Waterbirds on land). Models trained with ERM exhibit higher self-influence scores for minority subpopulations, suggesting increased memorization in these groups. In contrast, models trained with MAT show more uniform self-influence distributions across both majority and minority subpopulations. The rightmost plots display the proportion of samples in different self-influence intervals, with MAT producing a more balanced distribution compared to ERM. Further details can be found in Appendix D.

## 5    MEMORIZATION: THE GOOD, THE BAD, AND THE UGLY

In this work, we showed that the combination of memorization and spurious correlations, could be key reason for poor generalization. Neural networks can exploit spurious features and memorize exceptions to achieve zero training loss, thereby avoiding learning more generalizable patterns. However, an interesting and somewhat controversial question arises: *Is memorization always bad?*

To explore this, we look into a simple regression task to understand different types of memorization and their effects on generalization. We argue that the impact of memorization on generalization can vary depending on the nature of the data and the model's learning dynamics, and we categorize these types of memorization into three distinct forms. The task is defined as follows,

**Setup 5.1.** *Let $x_y \in \mathbb{R}$ be a scalar feature that determines the true target, $y^* = f(x_y)$. Let $\mathcal{D} = \{(\boldsymbol{x}_i, y_i)\}_{i=1}^n$ be a dataset consisting of input-target pairs $(\boldsymbol{x}, y)$. Define the input vector as*

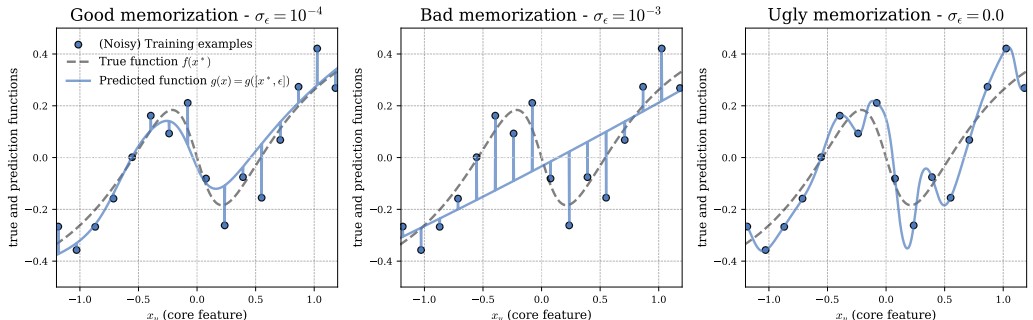

Figure 3: Three types of memorization in regression models trained with different levels of example-specific features ($\sigma_\epsilon$). The plots show the ERM-trained model $g(x) = g(x_y, \epsilon)$ (solid blue line) versus the true underlying function $f(x_y)$ (dashed gray line) and the noisy training examples. In all the three, the models are trained until the training loss goes below $10^{-6}$. **Good memorization (Left,** $\sigma_\epsilon = 10^{-4}$): Model learns the true function $f(x_y)$ well but slightly memorizes residual noise in the training data using the input example-specific features $\epsilon$. This type of memorization is benign, as it does not compromise generalization. **Bad memorization (Middle,** $\sigma_\epsilon = 10^{-3}$): The model relies more on example-specific features than learning the true function $f(x_y)$, leading to partial learning of $f(x_y)$ and fitting of noise-dominated input features. This type of memorization impedes learning of generalizable patterns and is considered malign. **Ugly memorization (Right,** $\sigma_\epsilon = 0.0$): Without example-specific features, the model overfits the training data, including label noise, resulting in a highly non-linear and complex model that fails to generalize to new data. This type is referred to as catastrophic overfitting.

$x = concat(x_y, \epsilon) \in \mathbb{R}^{d+1}$, *where* $\epsilon \sim \mathcal{N}(0, \sigma_\epsilon^2 I) \in \mathbb{R}^d$ *represents input example-specific features concatenated with the true feature* $x_y$. *The target is defined as* $y = y^* + \xi$, *where* $\xi \sim \mathcal{N}(0, \sigma_\xi)$ *represents additive target noise.*

In this context, $x_y$ can be interpreted as the core feature (e.g., the object in an object classification task), $\epsilon$ as irrelevant random example-specific features, and $\xi$ as labeling noise or error. Now, consider training linear regression models $\hat{y} = g(x)$ on this dataset. Fixing $\sigma_\xi$, we train three models under three different input example-specific features levels: $\sigma_\epsilon \in \{0, 10^{-4}, 10^{-3}\}$. The results, summarized in Figure 3, showcases three types of memorization:

**The Good: when memorization benefits generalization.** At an intermediate level of example-specific features, $\sigma_\epsilon = 10^{-4}$, the model effectively captures the true underlying function, $f(x_y)$. However, due to the label noise, the model cannot achieve a zero training loss solely by learning $f(x_y)$. As a result, it begins to memorize the residual noise in the training data by using the example-specific features $\epsilon$. This is evidenced by sharp spikes at each training point, where the model, $g(x)$, precisely predicts the noisy label if given the exact same input as during training. Nevertheless, for a neighboring test example with no example-specific features, the model's predictions align well with $f(x_y)$, demonstrating good generalization.

This phenomenon is often referred to as "benign overfitting" where a model can perfectly fit (overfit in fact) the training data while relying on example-specific features, yet still generalize well to unseen data (Belkin et al., 2019a; Muthukumar et al., 2020; Bartlett et al., 2020). The key insight is that the overfitting in this case is "benign" because the model's memorization by relying on example-specific features does not compromise the underlying structure of the true signal. Instead, the model retains a close approximation to the true function on test data, even though it memorizes specific noise in the training data. This has been shown to occur particularly in over-parameterized neural networks (Belkin et al., 2019b; Nakkiran et al., 2021).

**The Bad: when memorization prevents generalization.** At a higher level of example-specific features, $\sigma_\epsilon = 10^{-3}$, the model increasingly rely on thesefeatures $\epsilon$ rather than fully learning the true underlying function $f(x_y)$. In this case, memorization is more tempting for the model because the

example-specific features dominates the input, making it difficult to recover the true signal. As a result, the model $g(x)$ might achieve zero training loss by only partially learning $f(x_y)$ and instead relying heavily on the example-specific features in the inputs to fit the remaining variance in the training data.

This is an instance of bad memorization as it hinders the learning of generalizable patterns, the case we studied in this work. This phenomenon is referred to as "malign overfitting" in Wald et al. (2022), where a model fits the training data but in a way that compromises its ability to generalize, especially in situations where robustness, fairness, or invariance are critical.

It is important to note that both good and bad memorization stem from the same learning dynamics. ERM, and the SGD that drives it, do not differentiate between the types of correlations or features they are learning. Whether a features contributes to generalization or memorization is only revealed when the model is evaluated on held-out data. If the features learned are generalizable, the model will perform well on new data; if they are not, the model will struggle, showing its reliance on memorized, non-generalizable patterns.

**The Ugly: Catastrophic overfitting**  Finally, consider the case where there is no example-specific features, $\sigma_\epsilon = 0.0$. In this case, the model may initially capture the true function $f(x_y)$, but due to the presence of label noise, it cannot achieve zero training loss by learning only $f(x_y)$. Unlike the previous cases, the absence of example-specific features means the model has no additional features to leverage in explaining the residual error. As a result, the model is forced to learn a highly non-linear and complex function of the input $x = x_y$ to fit the noisy labels.

In this situation, memorization is ugly: The model may achieve perfect predictions on the training data, but this comes at the cost of catastrophic overfitting— where the model overfits so severely that it not only memorizes every detail of the training data, including noise, but also loses its ability to generalize to new data (Mallinar et al., 2022). Early stopping generally can prevent this type of severe overfitting.

## 6    RELATED WORK

**Detecting Spurious Correlations.**    Early methods for detecting spurious correlations rely on human annotations (Kim et al., 2019; Sagawa et al., 2019; Li & Vasconcelos, 2019), which are costly and susceptible to bias. Without explicit annotations, detecting spurious correlations requires assumptions. A common assumption is that spurious correlations are learned more quickly or are simpler to learn than core features (Geirhos et al., 2020; Arjovsky et al., 2019; Sagawa et al., 2020). Based on this, methods like Just Train Twice (JTT) (Liu et al., 2021), Environment Inference for Invariant Learning (EIIL) (Creager et al., 2021), Too-Good-To-Be-True Prior (Dagaev et al., 2023), and Correct-n-Contrast (CnC) (Zhang et al., 2022) train models with limited capacity to identify "hard" (minority) examples. Other methods such as Learning from Failure (LfF) (Nam et al., 2020) and Logit Correction (LC) (Liu et al., 2022) use generalized cross-entropy to bias classifiers toward spurious features. Closely related to this work is Cross-Risk Minimization (XRM) Pezeshki et al. (2023), where uses the held-out mistakes as a signal for the spurious correlations.

**Mitigating Spurious Correlations.**    Reweighting, resampling, and retraining techniques are widely used to enhance minority group performance by adjusting weights or sampling rates (Idrissi et al., 2022; Nagarajan et al., 2020; Ren et al., 2018). Methods like Deep Feature Reweighting (DFR) (Kirichenko et al., 2022) and Selective Last-Layer Finetuning (SELF) (LaBonte et al., 2024) retrain the last layer on balanced or selectively sampled data. Automatic Feature Reweighting (AFR) (Qiu et al., 2023) extends these methods by automatically upweighting poorly predicted examples without needing explicit group labels. GroupDRO (Sagawa et al., 2019) minimizes worst-case group loss, while approaches like LfF and JTT increase loss weights for likely minority examples. Data balancing can also be achieved through data synthesis, feature augmentation, or domain mixing (Hemmat et al., 2023; Yao et al., 2022; Han et al., 2022).

*Logit adjustment* methods are another line of work for robust classification under imbalanced data. Menon et al. (2020) propose a method that corrects model predictions based on class frequencies, building on prior work in post-hoc adjustments (Collell et al., 2016; Kim & Kim, 2020; Kang et al., 2019). Other methods, such as Label-Distribution-Aware Margin (LDAM) loss (Cao et al., 2019),

Balanced Softmax (Ren et al., 2020), Logit Correction (LC) (Liu et al., 2022), and Unsupervised Logit Adjustment (uLA) (Tsirigotis et al., 2024), adjust classifier margins to handle class or group imbalance effectively.

**Memorization**   Memorization in neural networks has received significant attention since the work of Zhang et al. (2021), which showed that these models can perfectly fit the training data, even with completely random labels. Several studies have studied the nuances of memorization across various scenarios (Zhang et al., 2019; Feldman, 2020; Feldman & Zhang, 2020; Brown et al., 2021; 2022; Anagnostidis et al., 2022; Garg et al., 2023; Attias et al., 2024). Feldman (2020); Feldman & Zhang (2020) examined how memorization contributes to improved performance on the tail of the distribution, especially on visually similar training and test examples, and highlighted the role of memorizing outliers and mislabeled examples in preserving generalization, akin to our concept of "good memorization" terminology (Section 5).

**Memorization and Spurious Correlations.**   Research has shown that memorization in neural networks can significantly affect model robustness and generalization. Arpit et al. (2017); Maini et al. (2022); Stephenson et al. (2021); Maini et al. (2023); Krueger et al. (2017) explore memorization's impact on neural networks, examining aspects like loss sensitivity, curvature, and the layer where memorization occurs. Yang et al. (2022) investigate "rare spurious correlations," which are akin to example-specific noise features that models memorize. Bombari & Mondelli (2024) provide a theoretical framework quantifying the memorization of spurious features, differentiating between model stability with respect to individual samples and alignment with spurious patterns. Finally, Yang et al. (2024) propose Residual-Memorization (ResMem), which combines neural networks with k-nearest neighbor-based regression to fit residuals, enhancing test performance across benchmarks.

## 7   CONCLUSION

In this work, we show that while spurious correlations are inherently problematic, they become particularly harmful when paired with memorization. This combination drives the training loss to zero too early, stopping learning before the model can capture more meaningful patterns. To address this, we propose Memorization-Aware Training (MAT), which leverages the negative effects of memorization to mitigate the influence of spurious correlations. Notably, we highlight that memorization is not always detrimental; its impact varies with the nature of the data. While MAT mitigates the negative effects of memorization in the presence of spurious correlations, there are cases where memorization can benefit generalization or even be essential (Feldman & Zhang, 2020). Future work could focus on distinguishing these scenarios and exploring the nuanced role of memorization in large language models (LLMs). Recent work (Carlini et al., 2022; Schwarzschild et al., 2024; Antoniades et al., 2024) have highlighted the importance of defining and understanding memorization in LLMs, as it can inform how these models balance between storing training data and learning generalizable patterns.

## ACKNOWLEDGEMENTS

We thank Kartik Ahuja, Andrei Nicolicioiu, and Reyhane Askari Hemmat for their invaluable feedback and discussions. RB acknowledges support from the Canada CIFAR AI Chair Program and the Canada Excellence Research Chairs (CERC) Program, as well as additional support from the UNIQUE scholarship. Part of the early experiments were conducted using computational resources provided by Mila Quebec AI Institute. Finally, we thank the FAIR leadership for their support in publishing this work.

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

# A  EXPERIMENTAL DETAILS

## A.1  EXPERIMENTS ON SUBPOPULATION SHIFT

For the results presented in Table 1, we follow the experimental settings in Pezeshki et al. (2023). The hyperparameter search involves testing 16 random hyperparameter combinations sampled from the search space described in Table 3, using a single random seed. We select the hyperparameter combination and the early-stopping iteration that achieve the highest validation worst-group accuracy, either with ground truth group annotations or pseudo annotations, depending on the method, or the worst-class accuracy if groups are not available. Then, to calculate the results with mean and standard deviation, we repeat the best-chosen hyperparameter experiment 10 times with different random seeds. Finally, to ensure a fair comparison between different methods, we always report the test worst-group accuracy based on the ground-truth (human annotated) group annotations provided by each dataset.

We tested our method on 4 standard datasets: two image datasets, Waterbirds (Sagawa et al., 2019) and CelebA (Liu et al., 2015), and two natural language datasets, MultiNLI (Williams et al., 2017) and CivilComments (Borkan et al., 2019). The configuration of each dataset is provided below. For CelebA, predictors map pixel intensities into a binary "blonde/not-blonde" label. No individual face characteristics, landmarks, keypoints, facial mapping, metadata, or any other information was used to train our CelebA predictors. We use a pre-trained ResNet-50 (He et al., 2016) for image datasets. For text datasets, we use a pre-trained BERT (Devlin et al., 2018). We initialized the weights of the linear layer added on top of the pre-trained model with zero. All image datasets have the same pre-processing scheme, which involves resizing and center-cropping to $224 \times 224$ pixels without any data augmentation. We use SGD with momentum of $0.9$ for the Waterbirds dataset, and we employ AdamW (Loshchilov & Hutter, 2017) with default values of $\beta_1 = 0.9$ and $\beta_2 = 0.999$ for the other datasets.

**Dataset.**  The detailed statistics for all datasets are provided in Table 2, together with the descriptions of each task below.

- Waterbirds (Sagawa et al., 2019): The Waterbirds dataset is a combination of the Caltech-UCSD Birds 200 dataset (Wah et al., 2011) and the Places dataset (Zhou et al., 2017). It consists of images where two types of birds (Waterbirds and Landbirds) are placed on either water or land backgrounds. The objective is to classify the type of bird as either "Waterbird" or "Landbird", with the background (water or land) introducing a spurious correlation.

- CelebA (Liu et al., 2015): The CelebA dataset is a binary classification task where the objective is to classify hair as either "Blond" or "Not-Blond", with gender considered a spurious correlation.

- MultiNLI (Williams et al., 2017): The MultiNLI dataset is a natural language inference task in which the objective is to determine whether the second sentence in a given pair is "entailed by", "neutral with", or "contradicts" the first sentence. The spurious correlation is the presence of negation words.

- CivilComments (Borkan et al., 2019): The CivilComments dataset is a natural language inference task in which the objective is to classify whether a sentence is "Toxic" or "Non-Toxic".

Table 2: Summary of datasets used, including their data types, the number of classes, the number of groups, and the total dataset size for each.

| Dataset | Data type | Num. of classes | Num. of groups | Train size |
|---|---|---|---|---|
| Waterbirds | Image | 2 | 2 | 4795 |
| CelebA | Image | 2 | 2 | 162770 |
| MultiNLI | Text | 3 | 6 | 206175 |
| CivilComments | Text | 2 | 8 | 269038 |

The baseline methods we compared our method with include ERM, GroupDRO (Sagawa et al., 2019), LfF (Nam et al., 2020), JTT (Liu et al., 2021), LC (Liu et al., 2022), uLA (Tsirigotis et al., 2024), AFR (Qiu et al., 2023), XRM+GroupDRO (Pezeshki et al., 2023). The results for ERM, GroupDRO are based on our own implementation, while the results for rest of the methods are adapted from the respective papers.

Table 3: Hyperparameter search space. ERM and MAT share the same hyperparameter search space, except that MAT has one additional hyperparameter, $\tau$, which is used in the softmax function as the temperature parameter to control the sharpness/smoothness of the output distribution.

| algorithm | hyper-parameter | ResNet | BERT |
|---|---|---|---|
| ERM & MAT | learning rate
weight decay
batch size
dropout | $10^{\text{Uniform}(-5,-3)}$
$10^{\text{Uniform}(-6,-3)}$
$2^{\text{Uniform}(5,7)}$
— | $10^{\text{Uniform}(-6,-4)}$
$10^{\text{Uniform}(-6,-3)}$
$2^{\text{Uniform}(4,6)}$
$\text{Random}([0, 0.1, 0.5])$ |
| MAT specific | $\tau$ | $\text{Random}([0.001, 0.01, 0.1])$ | $\text{Random}([0.001, 0.01, 0.1])$ |
| GroupDRO | $\eta$ | $10^{\text{Uniform}(-3,-1)}$ | $10^{\text{Uniform}(-3,-1)}$ |

**Methods.** We compared our method with a variety of baseline approaches presented in the following.

- ERM: Empirical Risk Minimization (ERM) trains a model by minimizing the average loss over the entire training dataset.

- GroupDRO (Sagawa et al., 2019): Group Distributionally Robust Optimization (GroupDRO) minimizes the worst-group loss across different predetermined groups in the training data using ground-truth group annotation.

- LfF (Nam et al., 2020): Learning from Failure (LfF) trains two models, one biased and one debiased. The biased model is trained to amplify reliance on spurious correlations using generalized cross-entropy, while the debiased model focuses on samples where the biased model fails.

- JTT (Liu et al., 2021): Just Train Twice (JTT) is a two-stage method that first identifies misclassified examples with ERM, then upweights them in the second stage to improve performance on hard-to-learn groups.

- LC (Liu et al., 2022): Logit Correction (LC) trains two models, one biased and one debiased. The biased model, trained with generalized cross-entropy, produces a correction term for the logits of the debiased model. Additionally, LC uses a Group MixUp strategy between minority and majority groups to further enrich the representation of the minority groups.

- uLA (Tsirigotis et al., 2024): Unsupervised Logit Adjustment (uLA) uses a pretrained self-supervised model to generate biased predictions, which are then used to adjust the logits of another model, improving robustness without needing group information.

- AFR (Qiu et al., 2023): Automatic Feature Reweighting (AFR) retrains the last layer of a model by up-weighting examples that the base model poorly predicted to reduce reliance on spurious features.

- XRM+GroupDRO (Pezeshki et al., 2023): Cross-Risk Minimization (XRM) automatically discovers environments in a dataset by training twin networks on disjoint parts of the training data that learn from each other's errors. XRM employs a label-flipping strategy to amplify model biases and better identify spurious correlations. In the next stage, an invariant learning method like GroupDRO uses the discovered environments (i.e., groups) to improve the worst group performance.

A.2 ANALYSIS OF MEMORIZATION SCORES

For the results presented in Figure 2, we used the TRAK framework (Park et al., 2023) to calculate self-influence scores for individual training data points. The TRAK framework computes the influence of each training point on a target dataset, generating an influence matrix, as illustrated in Figure 4. By setting the target dataset as the training dataset itself, we compute pairwise influence scores for all training data points, where the diagonal of this matrix represents the self-influence of each point on its own prediction. We visualized the distribution of these scores for both the majority and minority groups of the Waterbird dataset.

To utilize the framework, we provided the necessary model checkpoints saved during training. These checkpoints enabled TRAK to evaluate the contributions of specific training examples to the model's predictions.

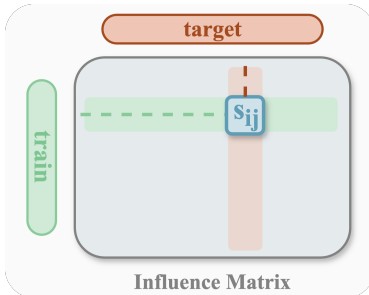

Figure 4: Influence Matrix: Influence scores of all the train and target data point pairs. Setting $target = train$ and $i = j$ reveals the self-influence.

# B EMPIRICAL INSIGHTS INTO THE EFFECTS OF REWEIGHTING AND LOGIT-SHIFTING

In this section, we empirically illustrate how gradient descent dynamics in logistic regression are influenced by different *weighting* and shifting schemes. ERM with uniform weights converges to the max-margin solution, aligning with the theoretical expectation (Soudry et al., 2018). Reweighting examples changes the optimization path but ultimately converges to the max-margin fixed point. In contrast, introducing a *shift* that scales with the logits changes the optimization landscape, leading to a distinct fixed point and a different classifier. These results highlight how both reweighting and logit-scaling shifts impact the intermediate learning dynamics, but only shifts fundamentally alter the convergence behavior. Below we provide details on the dataset, model, and experimental configurations, with results shown in Figure 5.

**Dataset.** The dataset consists of two-class synthetic data generated using Gaussian mixtures with overlapping clusters. We generate $n = 100$ samples in $\mathbb{R}^2$ with each class drawn from distinct Gaussian distributions. The features are standardized to zero mean and unit variance using a standard scaler, and the labels are assigned as $y \in \{-1, 1\}$.

**Model and Optimization.** The logistic regression model is parameterized as:

$$f(x; w, b) = Xw + b,$$

where $X \in \mathbb{R}^{n \times 2}$ is the input data, $w \in \mathbb{R}^2$ is the weight vector, and $b \in \mathbb{R}$ is the bias term. The gradient descent update is computed for the cross-entropy loss:

$$\mathcal{L} = -\frac{1}{n} \sum_{i=1}^{n} w_i \log \left( \frac{1}{1 + \exp(-y_i f(x_i; w, b))} \right),$$

where $w_i$ are per-example weights. We compare three configurations:

- **ERM:** Uniform weights are applied to all examples, i.e., $w_i = 1$ for all $i$.
- **Reweight:** We reduce the weights of examples for half of the examples of one class (e.g., $y = 1$) by setting $w_i = 0.1$ for these examples and $w_i = 1$ for the rest. This simulates scenarios where certain groups are more important during training, e.g. are minority.
- **Shift:** A margin-dependent shift is introduced by modifying the logits as:

$$f(x; w, b) = Xw + b + \delta_i \cdot \|w\|,$$

  where $\delta_i = 2$ for the same half of class $y = 1$ and $\delta_i = 0$ for the other. The scaling with the norm of the weight vector effectively adjusts the margin on each example.

To analyze the optimization dynamics, we plot the trajectories of the normalized weight vectors during training. Specifically, we plot: $w/\|w\| \times \log(t)$, where $t$ is the iteration step, to ensure the trajectories are visually distinguishable since all points would otherwise lie on the unit circle.

As shown in Figure 5, ERM converges to the max-margin solution. Reweighting changes the optimization trajectory initially but still converges to the max-margin fixed point. In contrast, the logit-shifting changes the fixed point of the optimization, making logit-shifting a more robust method and alleviating the need for precise early stopping.

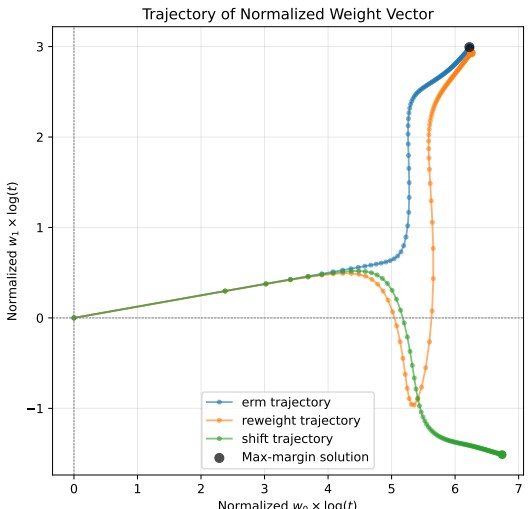

Figure 5: Weight vector trajectories for ERM (blue), reweighting (orange), and shifting (green) compared with the max-margin solution (black dot). Weight vector trajectories normalized as $w/\|w\| \times \log(t)$, where $t$ is the iteration step. The scaling by $\log(t)$ is done for better visualization only.

## C    FURTHER DISCUSSION ON EXPERIMENTAL RESULTS

Subpopulation shift methods operate under three setups regarding access to group annotations, each progressively increasing the task's difficulty.

The least restrictive scenario assumes full access to group annotations for both training and validation sets. This allows methods such as GroupDRO to leverage loss re-weighting with group-aware model selection and early stopping. While effective, this setup requires strong assumptions about the availability of group annotations. As noted in the original GroupDRO paper (Sagawa et al., 2020), achieving consistent performance in this setup often necessitates "stronger-than-typical $L_2$ regularization or early stopping". Without such measures, as shown in Sagawa et al. (2020) (Figure 2), GroupDRO's behavior can revert to ERM-like performance due to the implicit bias of gradient descent toward the max-margin solution.

A slightly more restrictive scenario assumes access to group annotations only for the validation set, which are used for *model selection* and *early stopping*. However, most algorithms, including GroupDRO, still rely on explicit group annotations during training, which must often be inferred using methods such as XRM. In contrast, MAT does not depend on training group annotations, making it more robust in cases where such information is unavailable.

The most restrictive and realistic scenario assumes no access to group annotations at any stage, either for training or validation. In this context, methods like MAT, which use per-example logits from held-out predictions, become particularly interesting. MAT operates without explicit group-specific information, making it flexible and robust in scenarios with ambiguous or undefined group structures. However, for the purpose of model selection, MAT does rely on explicit validation group annotations being inferred.

MAT's group-agnostic nature makes it particularly suitable for real-world scenarios where group definitions may be complex or unavailable. While GroupDRO relies on predefined or inferred group labels for its reweighting strategy, MAT uses per-example and soft adjustments without an explicit notion of group.

## D    ADDITIONAL INFLUENCE-SCORE EXPERIMENTS

In this section, we extend our experiments on influence-score analysis (refer to 4.2) across three methods: ERM, GroupDRO, and MAT. Figure 6 demonstrates that MAT effectively reduces self-influence scores relative to the other methods. Additionally, GroupDRO also shows some level reduction in the self-influence score of groups, specifically for the minority groups. This further indicates that memorization is limiting generalization, and mitigating it contributes to improved generalization in subpopulation shift settings.

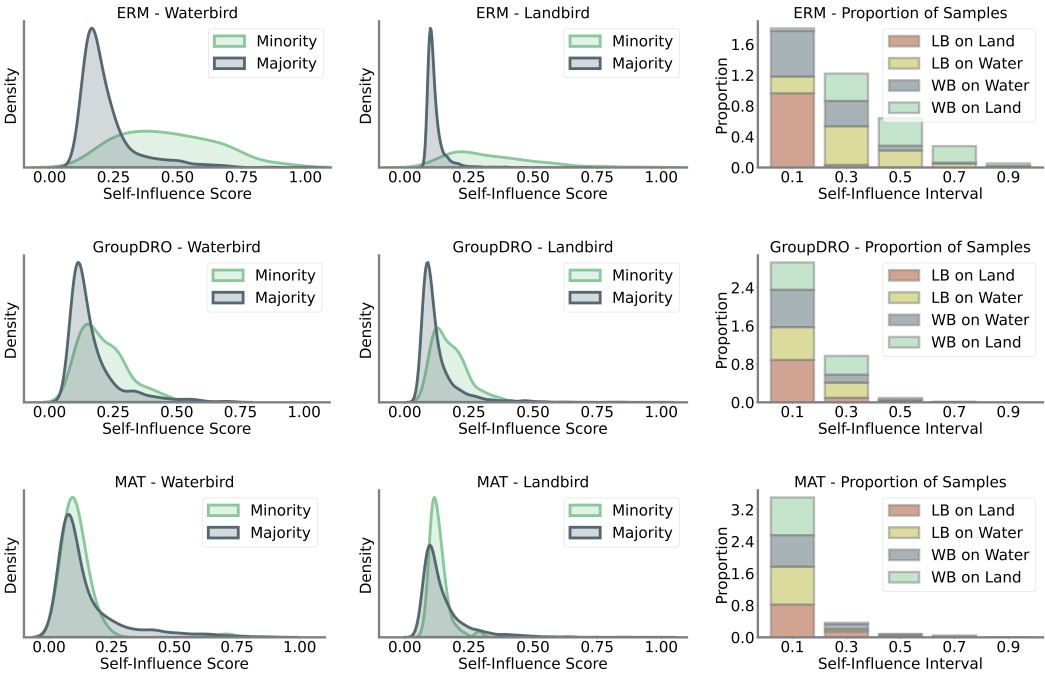

Figure 6: Self-Influence estimation of the Waterbird groups by ERM, GroupDRO and MAT. Each row corresponds to one of the training methods (ERM, GroupDRO, and MAT). The left and middle columns show the density distributions of self-influence scores for minority and majority subpopulations within each method, separately for Waterbirds and Landbirds. The right column depicts the proportion of samples across different self-influence score intervals for each subpopulation (LB on Land, LB on Water, WB on Water, WB on Land). The results illustrate that MAT achieves the most uniform and reduced self-influence distribution, indicating effective mitigation of memorization across subpopulations, especially in minority groups. GroupDRO also shows some reduction in self-influence scores for the critical group in this dataset, WB on Land.

## E    MULTIPLE SPURIOUS FEATURES SETUP

To test the capability of MAT in handling multiple spurious features, we have adapted our original setup in Section 2.1 and added a second spurious feature. Specifically, the data generation process is defined as:

$$\boldsymbol{x} = \begin{bmatrix} x_y \sim \mathcal{N}(y, \sigma_y^2) \\ x_{a_1} \sim \mathcal{N}(a, \sigma_a^2) \\ x_{a_2} \sim \mathcal{N}(a, \sigma_a^2) \\ \boldsymbol{\epsilon} \sim \mathcal{N}(0, \sigma_{\boldsymbol{\epsilon}}^2 \boldsymbol{I}) \end{bmatrix} \in \mathbb{R}^{d+2} \ \text{ where, } a = \begin{cases} y & \text{w.p. } \rho \\ -y & \text{w.p. } 1-\rho \end{cases} \text{ and } \rho = \begin{cases} \rho^{\text{tr}} & \text{(train)} \\ 0.5 & \text{(test)} \end{cases}.$$

**Illustrative Scenarios.** Similarly, we chose $\rho^{\text{tr}} = 0.9$ (i.e., the correlation of spurious features with labels) and $\gamma = 5$ (i.e., the strength of the spurious features) for both spurious features. In this new setup, with one main feature and two spurious features, there are four groups within each class, resulting in a total of eight groups. For simplicity in our visualization, we identified the group with the highest number of samples as the majority group and the group with the fewest samples as the minority group (i.e., the worst-performing group). Note that the number of samples in this minority group is even smaller than before, as the addition of the second spurious feature splits the previous minority group, introduced by the first spurious feature, into two even smaller groups.

Consistent with the single spurious feature setup (Section 2.1), Figure 7 shows that ERM, with memorization capacity, struggles to learn generalizable features and achieves only random-guess performance for the minority samples. In contrast, MAT effectively handles this scenario and nearly achieves perfect test accuracy for all groups.

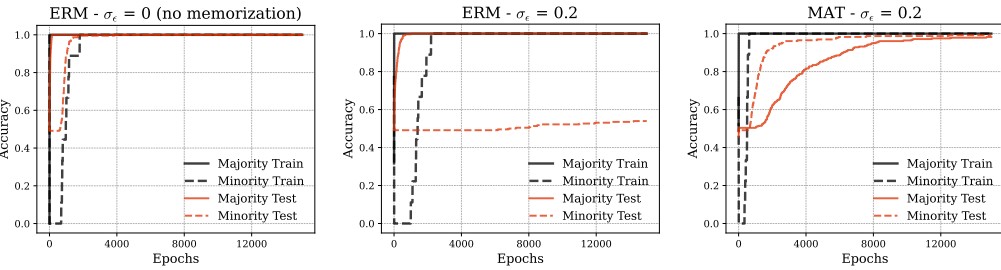

Figure 7: Similar to Figure 1, this figure illustrates the performance of ERM and MAT across different memorization setups in a scenario with **multiple spurious features**. The left panel shows ERM without memorization, where the model generalizes well to both majority and minority groups. The middle panel shows ERM with memorization capacity, where the model exhibits poor generalization for minority test samples. The right panel illustrates MAT, where it effectively learns invariant features and generalizes well to all groups.

## F Proofs

**Lemma F.1.** *Let* $p_1(y = j \mid \boldsymbol{x}) = \frac{e^{\phi_j(\boldsymbol{x})}}{\sum_{i=1}^{k} e^{\phi_i(\boldsymbol{x})}}$ *be a softmax over the logits* $\phi_j(\boldsymbol{x})$, *and define* $p_2(y = j \mid \boldsymbol{x})$ *such that* $p_2(y = j \mid \boldsymbol{x}) \propto w(j, \boldsymbol{x}) \cdot p_1(y = j \mid \boldsymbol{x})$ *for some weighting function* $w(j, \boldsymbol{x})$. *Then:*

$$p_2(y = j \mid \boldsymbol{x}) = \frac{e^{\phi_j(\boldsymbol{x}) + \log w(j, \boldsymbol{x})}}{\sum_{i=1}^{k} e^{\phi_i(\boldsymbol{x}) + \log w(i, \boldsymbol{x})}}.$$

*Proof.* Starting with the definition:

$$p_2(y = j \mid \boldsymbol{x}) \propto w(j, \boldsymbol{x}) \cdot p_1(y = j \mid \boldsymbol{x}).$$

Substituting the expression for $p_1(y = j \mid \boldsymbol{x})$:

$$p_2(y = j \mid \boldsymbol{x}) \propto w(j, \boldsymbol{x}) \cdot \frac{e^{\phi_j(\boldsymbol{x})}}{\sum_{i=1}^{k} e^{\phi_i(\boldsymbol{x})}}.$$

Since the denominator $\sum_{i=1}^{k} e^{\phi_i(\boldsymbol{x})}$ is constant with respect to $j$, it can be absorbed into the proportionality constant:

$$p_2(y = j \mid \boldsymbol{x}) \propto w(j, \boldsymbol{x}) \cdot e^{\phi_j(\boldsymbol{x})}.$$

Using the property $w(j, \boldsymbol{x}) \cdot e^{\phi_j(\boldsymbol{x})} = e^{\phi_j(\boldsymbol{x}) + \log w(j,\boldsymbol{x})}$, we have:

$$p_2(y = j \mid \boldsymbol{x}) \propto e^{\phi_j(\boldsymbol{x}) + \log w(j,\boldsymbol{x})}.$$

To obtain a valid probability distribution, we normalize by computing the normalization constant $Z$:

$$Z = \sum_{i=1}^{k} e^{\phi_i(\boldsymbol{x}) + \log w(i,\boldsymbol{x})}.$$

Therefore, the normalized $p_2(y = j \mid \boldsymbol{x})$ is:

$$p_2(y = j \mid \boldsymbol{x}) = \frac{e^{\phi_j(\boldsymbol{x}) + \log w(j,\boldsymbol{x})}}{Z} = \frac{e^{\phi_j(\boldsymbol{x}) + \log w(j,\boldsymbol{x})}}{\sum_{i=1}^{k} e^{\phi_i(\boldsymbol{x}) + \log w(i,\boldsymbol{x})}}.$$

This completes the proof. $\qquad\square$

## G  PROOF OF THEOREM 2.2

Setting the derivative of the objective function $\mathcal{L}^{\text{ERM}}$ (Equation equation 2) with respect to $w$ to zero gives the normal equation according to Appendix G.4

$$\frac{\partial \mathcal{L}^{\text{ERM}}}{\partial w} = \frac{1}{n} \sum_{j=1}^{n} (s(\boldsymbol{x}_i^{\top} w) - y_i)\boldsymbol{x}_i + \lambda w = 0,$$

where $s(\boldsymbol{x}_i^{\top} w) = \hat{p}^{\text{tr}}(y \mid \boldsymbol{x}_i; w)$, and solving for $w$ then gives

$$\widehat{w} = \sum_{i=1}^{n} \alpha_i \boldsymbol{x}_i,$$

with $\alpha_i := \frac{\pi_i - \widehat{\pi}_i}{\eta}$, with $\pi_i := 1_{\{y_i > 0\}}$, $\widehat{\pi}_i := s(v_i)$, $v_i := \boldsymbol{x}_i^{\top} \widehat{w}$, $\eta := n\lambda$.

Note that the $v_i$'s correspond to logits, while the $\alpha = (\alpha_1, \ldots, \alpha_n) \in \mathbb{R}^n$ should be thought of as the dual representation of the weights vector $\widehat{w}$. Indeed, by construction, one has

$$\widehat{w} = \boldsymbol{X}^{\top} \alpha, \tag{9}$$

where $\boldsymbol{X} \in \mathbb{R}^{n \times d}$ is the design matrix.

> **Notation.** Henceforth, with abuse of notation but WLOG, we will write $x_a = x_{spu} = a$ for the spurious feature, and therefore write $\boldsymbol{x}_i = (y_i, a_i, \epsilon_i)$, where $\epsilon_i$ are the example-specific noise features for the $i$ example and $y_i \in \{\pm 1\}$ is its label.

Our mission is then to derive necessary and sufficient conditions for $e > 0$, where

$$e := \gamma \widehat{w}_{spure} - \widehat{w}_{core} = \sum_{i=1}^{n} (\gamma^2 a_i - y_i)\alpha_i. \tag{10}$$

### G.1 FIXED-POINT EQUATIONS

Define subsets $I_\pm, S, L \subseteq [n]$ and integers $m, k \in [n]$ by

$$I_\pm := \{i \in [n] \mid y_i = \pm 1\}, \tag{11}$$
$$S := \{i \in [n] \mid a_i = \gamma y_i\}, \tag{12}$$
$$L := \{i \in [n] \mid a_i = -\gamma y_i\}, \tag{13}$$
$$m := \mathbb{E}\,|S| = pn, \quad k := \mathbb{E}\,|L| = (1-p)n. \tag{14}$$

Thus, $S$ (resp. $L$) corresponds to the sample indices in the majority (resp. the minority) class.

One computes the logits as follows

$$v_i = \boldsymbol{x}_i^\top \widehat{w} = \sum_{j=1}^n \alpha_j \boldsymbol{x}_j^\top \boldsymbol{x}_i = \sum_{j=1}^n \alpha_j y_j y_i + \gamma^2 \sum_{j=1}^n \alpha_j a_j a_i + \sum_{j=1}^n \alpha_j \epsilon_j^\top \epsilon_i$$

$$= \begin{cases} a + \sum_{j=1}^n \alpha_j \epsilon_j^\top \epsilon_i, & \text{if } i \in S \cap I_+, \\ b + \sum_{j=1}^n \alpha_j \epsilon_j^\top \epsilon_i, & \text{if } i \in S \cap I_-, \\ c + \sum_{j=1}^n \alpha_j \epsilon_j^\top \epsilon_i, & \text{if } i \in L \cap I_+, \\ e + \sum_{j=1}^n \alpha_j \epsilon_j^\top \epsilon_i, & \text{if } i \in L \cap I_-, \end{cases} \tag{15}$$

where $a, b, c, e \in \mathbb{R}$ are defined by

$$a := \gamma \widehat{w}_{spu} + \widehat{w}_{core} = \sum_{j=1}^n (\gamma^2 a_j + y_j)\alpha_j,$$

$$b := -\gamma \widehat{w}_{spu} - \widehat{w}_{core} = -\sum_{j=1}^n (\gamma^2 a_j + y_j)\alpha_j,$$

$$e := \gamma \widehat{w}_{spu} - \widehat{w}_{core} = \sum_{j=1}^n (\gamma^2 a_j - y_j)\alpha_j, \tag{16}$$

$$c := \widehat{w}_{core} - \gamma \widehat{w}_{spu} = \sum_{j=1}^n (-\gamma^2 a_j + y_j)\alpha_j.$$

Observe that

$$b = -a, \quad c = -e. \tag{17}$$

The following lemma will be crucial to our proof.

**Lemma G.1.** *If $a \geq 0$ and $e \geq 0$, then part 2 of Theorem 2.2 holds. On the other hand, if $a \geq 0$ and $e \leq 0$, then part 1 of Theorem 2.2 holds.*

*Proof.* Indeed, for a random test (i.e., held-out) point $(\boldsymbol{x}, a, y)$, we have

$$p(C_{ERM}(\boldsymbol{x}) = C_{spu}(\boldsymbol{x})) = p(\boldsymbol{x}_{spu} \times \boldsymbol{x}^\top \widehat{w} \geq 0)$$
$$= p(y\boldsymbol{x}_{spu}\widehat{w}_{core} + \gamma^2 \widehat{w}_{spu} + \boldsymbol{x}_{spu}\boldsymbol{x}_\epsilon^\top \widehat{w}_\epsilon \geq 0)$$
$$= p(-\boldsymbol{x}_{spu}\boldsymbol{x}_\epsilon^\top \widehat{w}_\epsilon \leq \gamma^2 \widehat{w}_{spu} + y\boldsymbol{x}_{spu}\widehat{w}_{core})$$

Now, independent of $y$, the random variable $-\boldsymbol{x}_{spu}\boldsymbol{x}_\epsilon^\top \widehat{w}$ has distribution $N(0, \sigma_\epsilon^2 \|\widehat{w}_\epsilon\|^2)$. Now, because $\widehat{w} = \boldsymbol{X}^\top \alpha$ by construction, the variance can be written as $\sigma_\epsilon^2 \|\widehat{w}_\epsilon\|^2 = \sigma_\epsilon^2 \|\boldsymbol{X}_\epsilon^\top \alpha\|^2$, which is itself chi-squared random variable which concentrates around its mean $\sigma_\epsilon^4 \|\alpha\|^2$. Furthermore, thanks to equation 21, $\|\alpha\|^2 \leq 1/(n\lambda^2)$, which vanishes in the limit

$$\lambda \to 0^+, \ n \to \infty, \ \lambda\sqrt{n} \to \infty. \tag{18}$$

We deduce that

$$p(C_{ERM}(\boldsymbol{x}) = C_{spu}(\boldsymbol{x})) \to p(\gamma^2 \widehat{w}_{spu} + y\boldsymbol{x}_{spu}\widehat{w}_{core} \geq 0)$$
$$= \rho 1_{\{\gamma \widehat{w}_{spu} + \widehat{w}_{core} \geq 0\}} + (1-\rho)1_{\{\gamma \widehat{w}_{spu} - \widehat{w}_{core} \geq 0\}}$$
$$= \rho 1_{\{a \geq 0\}} + (1-\rho)1_{\{e \geq 0\}}.$$

Thus, if $a \geq 0$ and $e \geq 0$, we must have $p(C_{ERM}(\boldsymbol{x}) = C_{spu}(\boldsymbol{x})) = \rho + 1 - \rho = 1$, that is, part 2 of Theorem 2.2 holds. In other words,

$$p\left(\widehat{y}_{\text{ERM}}^{\text{ho}}(\boldsymbol{x}) = a\right) \to 1.$$

On the other hand, one has

$$
\begin{aligned}
p(C_{ERM}(\boldsymbol{x}) = C_{core}(\boldsymbol{x})) &= p(\boldsymbol{x}_{core} \times \boldsymbol{x}^\top \widehat{w} \geq 0) = p(\widehat{w}_{core} + y \boldsymbol{x}_{spu} \widehat{w}_{spu} \geq 0) \\
&= q 1_{\{\widehat{w}_{core} + \gamma \widehat{w}_{spu} \geq 0\}} + (1 - q) 1_{\{\widehat{w}_{core} - \gamma \widehat{w}_{spu} \geq 0\}} \\
&= q 1_{\{a \geq 0\}} + (1 - q) 1_{\{e \leq 0\}},
\end{aligned}
$$

where $q := p(a = y)$.

We deduce that if $a \geq 0$ and $e \leq 0$, then $p(C_{ERM}(\boldsymbol{x}) = C_{core}(\boldsymbol{x})) = q + 1 - q = 1$, i.e part 2 of Theorem 2.2 holds. In other words,

$$p\left(\widehat{y}_{\text{ERM}}^{\text{ho}}(\boldsymbol{x}) = y\right) \to 1.$$

$\square$

## G.2 Structure of the Dual Weights

The following result shows that the dual weights $\alpha_1, \ldots, \alpha_n$ cluster into 4 lumps corresponding to the following 4 sets of indices $S \cap I_+$, $S \cap I_-$, $L \cap I_+$, and $L \cap I_-$.

**Lemma G.2.** *There exist positive constants $A, B, C, E > 0$ such that the following holds with large probability uniformly over all indices $i \in [n]$*

$$
\alpha_i \simeq
\begin{cases}
A, & \text{if } i \in S \cap I_+, \\
-B, & \text{if } i \in S \cap I_-, \\
C, & \text{if } i \in L \cap I_+, \\
-E, & \text{if } i \in L \cap I_-.
\end{cases}
\tag{19}
$$

*Furthermore, the empirical probabilities predicted by ERM are given by*

$$
\widehat{\pi}_i = 1_{\{y_i = 1\}} - \eta \alpha_i =
\begin{cases}
1 - \eta A, & \text{if } i \in S \cap I_+, \\
\eta B, & \text{if } i \in S \cap I_-, \\
1 - \eta C, & \text{if } i \in L \cap I_+, \\
\eta E, & \text{if } i \in L \cap I_-.
\end{cases}
\tag{20}
$$

*Proof.* First observe that

$$\|\alpha\| \leq \frac{1}{\lambda \sqrt{n}}. \tag{21}$$

Indeed, one computes

$$\|\alpha\|^2 = \frac{1}{\eta^2} \sum_{i=1}^n (\pi_i - \widehat{\pi}_i)^2 \leq \frac{1}{\eta^2} \sum_{i=1}^n 1 \leq \frac{n}{\eta^2} = \frac{1}{\lambda^2 n}.$$

Next, observe that $\sum_j \alpha_j \epsilon_j^\top \epsilon_i = \alpha_i \|\epsilon_i\|^2 + \sum_{j \neq i} \alpha_j \epsilon_j^\top \epsilon_i \simeq \sigma_\epsilon^2 \alpha_i d$. This is because $\alpha_i \|\epsilon_i\|^2$ concentrates around it mean which equals $\sigma_\epsilon^2 \alpha_i d$, while w.h.p,

$$\frac{1}{\sigma_\epsilon^2 d} \sup_{i \in [n]} \left| \sum_{j \neq i} \alpha_j \epsilon_j^\top \epsilon_i \right| \lesssim \|\alpha\| \sqrt{\frac{n \log n}{d}} = \sigma_\epsilon \|\alpha\| \sqrt{n} \cdot \sqrt{\frac{\log n}{d}} \leq \sigma_\epsilon \lambda \sqrt{\frac{\log n}{d}} = o(1).$$

The above is because $\lambda \to 0$ and $(\log n)/d \to 0$ by assumption. Henceforth we simply ignore the contributions of the terms $\sum_{j \neq i} \alpha_j \epsilon_j^\top \epsilon_i$. We get the following equations in the limit 18

$$
v_i = \begin{cases}
\sigma_\epsilon^2 \alpha_i d + a, & \text{if } i \in S \cap I_+, \\
\sigma_\epsilon^2 \alpha_i d + b, & \text{if } i \in S \cap I_-, \\
\sigma_\epsilon^2 \alpha_i d + c, & \text{if } i \in L \cap I_+, \\
\sigma_\epsilon^2 \alpha_i d + e, & \text{if } i \in L \cap I_-,
\end{cases}
$$

$$
\eta \alpha_i = y_i - s(v_i) = \begin{cases}
1 - s(\sigma_\epsilon^2 \alpha_i d + a), & \text{if } i \in S \cap I_+, \\
-s(\sigma_\epsilon^2 \alpha_i d + b), & \text{if } i \in S \cap I_-, \\
1 - s(\sigma_\epsilon^2 \alpha_i d + c), & \text{if } i \in L \cap I_+, \\
-s(\sigma_\epsilon^2 \alpha_i d + e), & \text{if } i \in L \cap I_-.
\end{cases}
\tag{22}
$$

Now, because of monotonicity of $\sigma$, we can find $A, B, C, E > 0$ such that

$$
\alpha_i = \begin{cases}
A, & \text{if } i \in S \cap I_+, \\
-B, & \text{if } i \in S \cap I_-, \\
C, & \text{if } i \in L \cap I_+, \\
-E, & \text{if } i \in L \cap I_-,
\end{cases}
$$

as claimed. $\qquad \square$

We will make use of the following lemma.

**Lemma G.3.** *In the unregularized limit $\lambda \to 0^+$, it holds that $\eta A, \eta B, \eta C, \eta E \in [0, 1/2]$.*

*Proof.* Indeed, in that unregularized limit, ERM attains zero classification error on the training dataset (part 0 of Theorem 2.2). This means mean that $\hat{\pi}_i \geq 1/2$ iff $y_i = 1$, and the result follows. $\qquad \square$

### G.3 FINAL TOUCH (PROOF OF THEOREM 2.2)

We resume the proof of Theorem 2.2. The scalars $A, B, C, E$ must verify

$$
\begin{aligned}
\eta A &= 1 - s(\sigma_\epsilon^2 A d + a) = s(-\sigma_\epsilon^2 A d - a), \\
\eta B &= s(-\sigma_\epsilon^2 B d + b) = s(-\sigma_\epsilon^2 B d - a) = 1 - s(\sigma_\epsilon^2 B d + a), \\
\eta E &= s(-\sigma_\epsilon^2 E d + e), \\
\eta C &= 1 - s(\sigma_\epsilon^2 C d + c) = 1 - s(\sigma_\epsilon^2 C d - e) = s(-\sigma_\epsilon^2 C d + e).
\end{aligned}
\tag{23}
$$

We deduce that

$$
A = B, \quad C = E, \tag{24}
$$

$$
\eta A = s(-\sigma_\epsilon^2 A d - a), \quad \eta E = s(-\sigma_\epsilon^2 E d + e). \tag{25}
$$

**Proof of Part 1 of Theorem 2.2.** In particular, for the case with no example-specific features where $\sigma_\epsilon \to 0^+$, we have $\eta A \simeq s(-a)$ and $\eta E \simeq s(e)$. We know from Lemma G.3 that $\eta A, \eta E \leq 1/2$. This implies $a \geq 0$ and $e \leq 0$, and thanks to Lemma G.1, we deduce part 1 of Theorem 2.2

**Proof of Part 2 of Theorem 2.2.** In remains to show that $a \geq 0$ and $e \geq 0$ in the noisy regime $\sigma_\epsilon > 0$, and then conclude via Lemma G.1.

Define $N_1 := |S \cap I_+|$, $N_2 := |S \cap I_-|$, $N_3 := |L \cap I_+|$, $N_4 := |L \cap I_-|$. Note that from the definition of $a, b, c, e$ in equation 16, one has

$$
\begin{aligned}
a &= (\gamma^2 + 1)(N_1 + N_2)A - (\gamma^2 - 1)(N_3 + N_4)E, \\
e &= (\gamma^2 - 1)(N_1 + N_2)A - (\gamma^2 + 1)(N_3 + N_4)E, \\
b &= -a, \quad c = -e, \\
\eta A &= s(-\sigma_\epsilon^2 A d - a), \quad \eta E = s(-\sigma_\epsilon^2 E d + e), \\
B &= A, \quad C = E.
\end{aligned}
\tag{26}
$$

We now show that $a \geq 0$ and $e \geq 0$ under the conditions $d \gg \log n$ and $\gamma \gg \sigma_\epsilon \sqrt{d/m}$.

Indeed, under the second condition, the following holds w.h.p

$$\sigma_\epsilon^2 d + (\gamma^2 + 1)(N_1 + N_2) = ((\gamma^2 + 1)(N_1 + N_2) + \sigma_\epsilon^2 d) \simeq ((\gamma^2 + 1)m + \sigma_\epsilon^2 d)$$
$$\simeq (\gamma^2 + 1)m \simeq (\gamma^2 + 1)(N_1 + N_2),$$

where we have used the fact that $N_1 + N_2$ concentrates around its mean $m = pn$.

We deduce that

$$\sigma_\epsilon^2 Ad + a = (\sigma_\epsilon^2 d + (\gamma^2 + 1)(N_1 + N_2))A - (\gamma^2 - 1)(N_3 + N_4)E$$
$$\simeq (\gamma^2 + 1)(N_1 + N_2)A - (\gamma^2 - 1)(N_3 + N_4)E$$
$$\simeq a,$$

from which we get.

$$1/2 \geq \eta A \geq s(-\sigma_\epsilon^2 Ad - a) = s(-(1 + o(1))a) = s(-a) + o(1),$$

i.e $s(-a) \geq 1/2 - o(1)$. But this can only happen if $a \geq 0$.

Finally, the conditions $d \gg \log n$ and $\gamma \gg \sigma_\epsilon \sqrt{d/m}$ imply $\gamma \gg K\sigma_\epsilon \sqrt{d/k}$ and $g \geq K \log(3n)$ for any constant $K > 0$. Theorem 1 of Puli et al. (2023) then gives $e = \gamma \widehat{w}_{spu} - \widehat{w}_{core} > 0$, and we are done. □

### G.4 DERIVATIVE OF THE OBJECTIVE FUNCTION

Given the objective function (Equation equation 2):

$$\mathcal{L}^{\text{ERM}} = \frac{1}{n} \sum_{i=1}^n l(y_i, \hat{p}^{\text{tr}}(y \mid \boldsymbol{x}_i; w)) + \frac{\lambda}{2}||\boldsymbol{w}||^2,$$

where $l(y_i, \hat{p}^{\text{tr}}(y \mid \boldsymbol{x}_i; w))$ is the cross-entropy loss for binary classification, we can write:

$$l(y_i, \hat{p}^{\text{tr}}(y \mid \boldsymbol{x}_i; w)) = -y_i \log(\hat{p}^{\text{tr}}(y_i \mid \boldsymbol{x}_i; w)) - (1 - y_i) \log(1 - \hat{p}^{\text{tr}}(y_i \mid \boldsymbol{x}_i; w)),$$

where $\hat{p}^{\text{tr}}(y \mid \boldsymbol{x}_i; w) = s(\boldsymbol{x}_i^\top w)$ is the predicted probability of class $y = 1$, and $s(\cdot)$ is the sigmoid function:

$$s(\boldsymbol{x}_i^\top w) = \frac{1}{1 + e^{-\boldsymbol{x}_i^\top w}}.$$

Now, the derivative of the $\mathcal{L}^{\text{ERM}}$ w.r.t. $w$ consists of two parts:

1. Derivative of the cross-entropy loss: we compute the derivative of the cross-entropy term w.r.t. $w$:

$$\frac{\partial}{\partial w} l(y_i, s(\boldsymbol{x}_i^\top w)) = \frac{\partial}{\partial w} \left[ -y_i \log(s(\boldsymbol{x}_i^\top w)) - (1 - y_i) \log(1 - s(\boldsymbol{x}_i^\top w)) \right].$$

Let $p_i = s(\boldsymbol{x}_i^\top w)$. Using the chain rule, we first calculate the derivative of the cross-entropy loss w.r.t. $p_i$:

$$\frac{\partial l(y_i, p_i)}{\partial p_i} = -\frac{y_i}{p_i} + \frac{1 - y_i}{1 - p_i}.$$

Next, we compute the derivative of $p_i = s(\boldsymbol{x}_i^\top w)$ w.r.t. $w$:

$$\frac{\partial p_i}{\partial w} = s(\boldsymbol{x}_i^\top w)(1 - s(\boldsymbol{x}_i^\top w))\boldsymbol{x}_i = p_i(1 - p_i)\boldsymbol{x}_i.$$

Using the chain rule:

$$\frac{\partial l(y_i, s(\boldsymbol{x}_i^\top w))}{\partial w} = \left( -\frac{y_i}{s(\boldsymbol{x}_i^\top w)} + \frac{1 - y_i}{1 - s(\boldsymbol{x}_i^\top w)} \right) \cdot s(\boldsymbol{x}_i^\top w)(1 - s(\boldsymbol{x}_i^\top w))\boldsymbol{x}_i.$$

Simplifying the expression:

$$\frac{\partial l(y_i, s(\boldsymbol{x}_i^\top w))}{\partial w} = (s(\boldsymbol{x}_i^\top w) - y_i)\boldsymbol{x}_i.$$

Thus, the derivative of the cross-entropy loss term is:

$$\frac{\partial}{\partial w}\left(\frac{1}{n}\sum_{i=1}^{n} l(y_i, \hat{p}^{\text{tr}}(y \mid \boldsymbol{x}_i; w))\right) = \frac{1}{n}\sum_{i=1}^{n}(s(\boldsymbol{x}_i^\top w) - y_i)\boldsymbol{x}_i.$$

2. Derivative of the regularization term: the second part of the objective function is the $\ell_2$-regularization term:

$$\frac{\lambda}{2}||\boldsymbol{w}||^2.$$

The derivative of this term w.r.t. $w$ is straightforward:

$$\frac{\partial}{\partial w}\left(\frac{\lambda}{2}||\boldsymbol{w}||^2\right) = \lambda w.$$

Final derivative: combining both terms, the derivative of the objective function $\mathcal{L}^{\text{ERM}}$ w.r.t. $w$ is:

$$\frac{\partial \mathcal{L}^{\text{ERM}}}{\partial w} = \frac{1}{n}\sum_{i=1}^{n}(s(\boldsymbol{x}_i^\top w) - y_i)\boldsymbol{x}_i + \lambda w.$$

This is the required derivative of the regularized cross-entropy loss function with respect to $w$.

