# OpenReview forum: "The Pitfalls of Memorization: When Memorization Hurts Generalization"
_ICLR.cc/2025/Conference — ICLR 2025 Poster_

### Official Review · Reviewer_ttKz · 2024-10-29

**Soundness:** 3
**Presentation:** 2
**Contribution:** 3
**Rating:** 6
**Confidence:** 4

**Summary:**

This paper studies the interplay between learning with spurious features and memorizing the noises in the training data. It shows that the combination of the two can be harmful for model generalization because the model lacks further incentive to learn the actual underlying generalizable solution. Based on this, the paper proposed a training paradigm called memorization-aware training (MAT) by utilizing a model trained with heldout data to shift the current model's logits.

**Strengths:**

1. This paper presents a study of how memorization impact generalization, with a specific focus on the presence of spurious features and distribution shift between training and testing.
2. The intuition is presented with both a synthetic data example and a theoretical construction.
3. A new method is proposed based on the intuition that perform competitively compared to the previous baselines.

**Weaknesses:**

1. Spurious features and distribution shift is at the core of the most of the paper. I believe it is better to reflect this in the paper title, which currently is framed quite broadly.
2. The presentation of the paper could be improved. For example

    1. The synthetic example and discussions in Section 5, while interesting, feels disconnected and unrelated to the rest of the paper. Does it mean the memorization at the presence of spurious feature are always good / bad / ugly type of memorization? Or is the proposed training method going to help mitigate any type of memorization discussed here? Can those types of memorization be easily identified in real world scenarios?

    2. Section 4.1 ends abruptly with a reference to Table 1 and a list of the baseline methods. Having some discussions of the results would be good since the proposed MAT algorithm is an important contribution of the paper. Right now looking at Table 1 it seems the proposed method is not outperforming previous methods when the group annotations for the validation data is available.

3. The logits shift is estimated with Eq (4) when annotation is not available. It would be great to have an ablation study of the accuracy of such estimation in one of the experiments.

4. The explanation of learning spurious feature + memorizing noise is interesting. However, it is still unclear from the results of the paper if this is what is happening in more general setting (i.e. without explicit spurious features and subpopulation shift). So it would be great to have some results on other standard ML benchmark as well.

5. The analysis in Section 4.2 show that the proposed method reduces self influence in the minority subpopulations. It would be great to include the same analysis on some baseline methods compared in Section 4.1 as well.

**Questions:**

1. In the experiments, does the validation set used to shift the model logits contain spurious feature?
2. L160: the paper says "the model *first* learns $x_a$ ... Once the model achieve nearly perfect accuracy on the majority examples, it *starts* to learn the minority examples". Could you explain where do we observe such order of learning from Figure 1?


==============
Post rebuttal: Thanks the authors for the response, especially the additional results and analysis. I've raised my rating.

---

> ### Author Response · Authors · 2024-11-23
>
> Thank you for your detailed review and constructive feedback. Below we address your questions. We are happy to continue the discussion in case of further follow-up questions. We have split our response into two parts for better readability.
>
> ## Memorization Types in Section 5:
>
> In Section 5, towards the end of the paper, we wanted to provide a thought-provoking discussion on the other side of memorization. As explicitly posed in line 377, we ask the question, *“Is memorization always bad?”* We apologize if our response to this question was unclear. The short answer is *no*, memorization is a double-edged sword. While it can sometimes harm generalization, there are situations where it plays a positive role, and ERM might even outperform methods like MAT in such cases.
>
> To address your specific questions:
>
> > Does it mean the memorization at the presence of spurious features is always good/bad/ugly?
>
> Memorization can indeed be categorized as good, bad, or ugly depending on the context, the inductive biases of the learning algorithm, and the very definition of spurious features (e.g., the type of leaves an insect is sitting on is spuriously correlated with the type of bug, yet entomologists use this signal to identify the insect). In this section, we only scratched the surface of this complex topic, hoping to inspire future research to disentangle the interplay between memorization and generalization more deeply.
>
> > Is the proposed training method going to help mitigate any type of memorization discussed here?
>
> As stated in line 439, MAT specifically addresses “bad memorization”. In scenarios where memorization is not harmful, or is even beneficial, MAT may hurt the performance. To see this, take the Waterbirds task: suppose that the task were to classify the background rather than the bird type. ERM would perform better by memorizing the minority (which are now considered mislabeled), while MAT would (incorrectly) learn the (now spurious) foreground feature.
>
> > Can those types of memorization be easily identified in real-world scenarios?
>
> As discussed in lines 60–63, the primary signal for identifying harmful memorization is the model's test performance, particularly on worst-group. This true generalization signal can guide us in determining which method to use. For standard tasks where harmful memorization is unlikely, ERM is generally sufficient. However, in scenarios where spurious correlations are anticipated, or when test performance suggests that, methods like MAT or other invariant learning techniques should be considered.
>
> We will revise the manuscript to make these points more explicit.

---

> ### Author Response · Authors · 2024-11-23
>
> ## Results Discussion about Table 1:
>
> Thank you for raising this point. We have now added a new section in Appendix E (highlighted in green for better visibility) titled "Further Discussion on Experimental Results", to provide more details. In the final version, we will move parts of it to the main text.
>
> The newly added section clarifies the different levels of assumptions about access to group annotations and highlights their implications for methods like MAT and GroupDRO. Specifically, we discuss how:
>
> - GroupDRO relies heavily on group annotations, either predefined or inferred, for its reweighting strategy and performs well when full or partial access to group annotations is available.
> - MAT, by contrast, is group-agnostic, requiring no explicit group annotations during training. This makes it more robust in scenarios where such annotations are unavailable or ambiguous, although it does rely on inferred validation group annotations for model selection.
> - We also emphasize that MAT’s ability to perform competitively without relying on training group annotations is a key strength, especially in real-world applications where such information is often unavailable.
>
>
> ## Ablation on Logits Shift Estimation:
> Thank you for the suggestion. In the newly added section in Appnedix F (highlighted in green for better visibility) titled "Ablation Study on Estimations in $p_a(y \mid x_i)$", we have now conducted an ablation study to evaluate the accuracy and impact of the components used in the logit shift estimation $p(a \mid \textbf{x}_i)$ and $p(y \mid a)$. Specifically, we compare configurations where these components are either replaced by their ground-truth values or estimated using Equations (4) and (5). The results, detailed in the newly added table in the paper, show that while using ground-truth values yields the best performance, the fully estimated version of MAT still significantly outperforms ERM. This highlights the robustness of the estimation process and demonstrates MAT’s effectiveness even without access to ground-truth annotations. We hope this addresses your question.
>
>
> ## Further Self-influence Analysis:
>
> Thank you for the suggestion. In the revised manuscript, we have added a self-influence analysis for GroupDRO in Appendix G (highlighted in green for better visibility) titled "Additional Influence-Score Experiments," to provide a direct comparison. The results confirm that MAT is more effective at reducing self-influence scores across all subpopulations (groups). GroupDRO also shows some level reduction in the self-influence score of groups, specifically for the minority groups. This further indicates that memorization is limiting generalization, and mitigating it contributes to improved generalization in subpopulation shift settings.
>
>
> ## Validation Set and the Amount Shift:
>
> To clarify, "MAT **does not** use the validation sets to shift the logits". MAT uses the held-out predections from the training set. These held-out predictions are obtained by randomly splitting the training data into two halves and training two models simultaneously. Consequently, each training sample is seen and trained on by only one model, while the other model provides the held-out predictions for it. This enables us to compute held-out predictions, which serve as a strong indicator of where models may fail, enabling logit shifting for each individual training sample. We will clarify this in the paper.
>
> ## Learning Order of Features:
>
> In Figure 1 (left and middle panels), you can observe that both the training and test accuracy for the **majority groups** (red and black solid lines) increase rapidly to nearly 100% accuracy after a few epochs. These majority groups are where the **spurious features $x_a$** dominate the learning dynamics, meaning the model first learns $x_a$. Meanwhile, we observe a delayed learning process (at epoch ~200) for the minority groups (red and black dashed line).
>
> In the left panel, where noise features are off ($\sigma_{\epsilon} = 0$) and memorization through these features is not possible, the model eventually learns the generalizable pattern (by learning the core feature $x_y$ from the minority groups, achieving 100% accuracy in **both** training and test sets for the minority groups.
>
> In contrast, in the middle panel, where noise features are present ($\sigma_{\epsilon} = 1$), memorization is possible. Here, the model tends to rely on these features instead of the core feature $x_y$ to reduce loss, resulting in 100% training accuracy, but significantly **lower test accuracy**.
>
>
> ---
> We’d be happy to clarify any remaining questions or engage further in discussion. Thanks again for your review and consideration!

---

> ### Author Response · Authors · 2024-11-25
> **Follow-Up on Rebuttal Review and Clarifications**
>
> Dear Reviewer,
>
> Thank you once again for taking the time to provide your valuable feedback on our paper. With the discussion phase nearing its end, and to ensure that our responses are fully considered, we request that you kindly confirm whether you’ve had a chance to review them.
>
> We hope our responses have addressed your concerns and clarified any ambiguities. If there are any further questions or suggestions, we would be glad to address them promptly.
>
> Additionally, if our responses have adequately resolved your concerns, we kindly ask you to consider reflecting this in your review score.
>
> Thank you!

---

> > ### Author Response · Authors · 2024-12-01
> > **Any Final Thoughts?**
> >
> > Dear reviewer, could you please let us know if you’ve had a chance to review our responses? We put a lot of effort into addressing your concerns. Thanks!

---

### Official Review · Reviewer_USfi · 2024-10-31

**Soundness:** 2
**Presentation:** 2
**Contribution:** 2
**Rating:** 6
**Confidence:** 2

**Summary:**

This paper examines the relationship between spurious correlations and memorization. A model trained with Empirical Risk Minimization (ERM), which has learned spurious correlations and has memorized irrelevant patterns, can have poor generalization. Spurious correlations result from patterns within the input data that are closely associated with the output, but not necessarily predictive (e.g., grass in a scene with cows). Memorization occurs when a model memorizes specific patterns rather than learning robust features that generalize to new examples.

To address this, the authors propose a novel approach, Memorization-Aware Training (MAT), a method that modifies the logits of the cross-entropy loss to discourage the model from relying on spurious features ("the indirect path"). By minimizing the log probability of the "indirect path", where the output y depends on a spurious feature a, MAT encourages the model to learn more generalizable patterns.

The accuracy of MAT is compared to various baseline methods under different sub-population label settings. Additionally, the paper shows that MAT effectively shifts the self-influence distribution, reducing the reliance on spurious correlations.

The authors additionally provide a thorough theoretical analysis and a detailed description MAT algorithm.

**Strengths:**

1. I really appreciated the first paragraph of the introduction and how it related to your work. This was very creative and smart, and helps the reader in understanding.
2. I like some the experiments in the paper and I have questions about others. First, for Figure 1, setting gamma to 5 creates a type of "worst-case" scenario with respect to spurious features. MAT is able to overcome this learning a generalizable function in the presence of noise. The experiments in Figure 2 show an improved self-influence score distribution with the use of MAT, which is a convincing way to present the effectiveness of your method.

**Weaknesses:**

1. One thing that the paper is missing is arguments as to why MAT is a better approach than other invariant methods? Or, alternatively, why should one use MAT over these other methods? What are the advantages and disadvantages, aside from sub-population labelling? The average accuracies among these methods seem to be similar. That is, there is no consistent 'best' approach. This doesn't invalidate the results, but I would like to know why and when I should choose MAT over another approach.
2. I think that this paper would benefit from a more realistic scenario where noise modifies the input features. The concatenation of noise to the input is a simplification, which serves a clear purpose; however, it is not representative of real-world noise, in which the input features are directly modified.
3. The rightmost image in Figure 3, with noisy labels, confused me in the sense that I don't understand how it demonstrates the benefit of using MAT.

**Questions:**

1. (Continuation of weakness #2) Why did you chose to concatenate the noise to the input, rather than add it to the input? How do you think your results would differ if you added the noise to the input, simulating noisy real-world data? Your results, which rely on this simplification, may not hold true to real-world tasks.
2. (Confusion) Looking at the results depicted in Figure 2, I noted that when training with ERM, the Waterbird on Water has a right-tailed self-influence score (e.g., in the right most figure, WB on Water has a large proportion of samples with a self-influence score of 0.3, yet LB on Land has a very tiny proportion of samples with a self-influence score of 0.3). Do you have any ideas on why this is the case? Is this due to a characteristic of the dataset? Of the sub-population images?
3. You can also address any weaknesses.

---

> ### Author Response · Authors · 2024-11-23
>
> Thank you for your thoughtful feedback and kind words about the introduction and experiments! Below we address your remarks and provide further contexts. We have split our response into three parts for better readability.
>
> ## Why and when should MAT be used?
>
> This is a valid question. As you pointed out, one major challenge in subpopulation shift problems is *group annotations*, which face two main issues: (1) Group annotation could be expensive, and (2) The second issue, which is somewhat hidden in this setup, is defining the groups themselves. In many real-world problems, the types and number of groups may not be easily defined. While datasets like Waterbirds have clear group structures (e.g., [Waterbird, Landbird] x [Water, Land]), many real-world scenarios lack such clarity.
>
> One MAT’s advantage is that it is *group-agnostic*, which eliminates the need for predefined or inferred group labels. Unlike GroupDRO, which depends on explicit group annotations (human-labeled or XRM-inferred), MAT uses soft held-out predictions for **individual examples**, enabling it to train without any notion of group. While MAT’s performance on current benchmarks is comparable to GroupDRO with XRM-inferred annotations, its group-agnostic nature holds promise for more complex, less structured setups, a direction we leave for future work.
>
> Another advantage of MAT is its reduced sensitivity to *early stopping* or *model selection*. Unlike reweighting methods such as GroupDRO, which converge to the same *max-margin solution* as ERM unless carefully regularized or stopped early, MAT fundamentally changes both the optimization trajectory and the final solution. This topic is discussed further in the newly added section in **Appendix D** (highlighted in yellow for better visibility) titled "Empirical Insights into the Effects of Reweighting and Logit-Shifting".
>
> That being said, as discussed in the paper, we only expect MAT to excels in setups where memorization is "bad". When memorization is good, even ERM may outperform MAT and any invariant learning method. For example, in the Waterbirds dataset, if the task were *background* classification, ERM’s reliance on background features would be advantageous, whereas MAT might learn a non-generalizable pattern (the foreground in this case).

---

> ### Author Response · Authors · 2024-11-23
>
> ## On the Role of Noise Dimensions:
>
> Thanks for pointing this out! We would like to clarify that the noise features we add to the inputs play a different role from their traditional usage, where input noise is employed as a form of regularization (e.g., Bishop, 1995 [Training with Noise is Equivalent to Tikhonov Regularization]). In our setup, these features are used to model any feature that, in expectation (given infinite data), is independent (and thus decorrelated) from the target labels. However, due to the finite sample size, they exhibit apparent empirical correlations and can be leveraged for memorization. We believe our setup closely resembles real-world scenarios, and we will illustrate this similarity by examining one of the datasets in our study: the Waterbird dataset.
>
> In our setup (Section 2.1), we define three types of features:
>
> 1. **Spurious feature:** Partially correlated with the label; learning it doesn't generalize to perfect accuracy in testing.
> 2. **Core feature:** Fully correlated with the label; learning it ensures perfect accuracy in testing.
> 3. **Noise features:** Uncorrelated with the label in expectation but may show empirical correlations in finite data, e.g., irrelevant pixels in an image.
>
> Now consider the Waterbird dataset. In the generation process of this dataset, we have two features: the foreground (the type of bird, such as "waterbird" or "landbird") and the background ("water" or "land"). The type of bird is considered the core feature because it is 100% correlated with the target label, while the background is considered the spurious feature, as it is not fully correlated. For instance, a small fraction of the "waterbirds" appear with a "land" background.
>
> However, in high-dimensional settings such as images, this distinction becomes less clear. Essentially, every pixel in an image dataset can be considered a feature, allowing a model to exploit any arbitrary combination of pixels for shortcut learning. For instance, a random pixel at position (x, y) might be distinguishable enough for the model to memorize a specific sample. Therefore, similar to our setup, such features can be considered noise features (also referred to as example-specific features): they are not correlated with the label in expectation but may be used to reduce the training loss for specific samples through memorization.
>
> Additionally, although we concatenate the noise feature to the inputs for simplicity, once the data passes through the network, all features (including the noise) interact and become mixed together in the representations. The network does not treat the noise as separate. For this reason, if we rotate the features (e.g., by multiplying them by an orthogonal matrix) and then add the noise, the results and conclusions remain unchanged.
>
> Lastly, we plan to rename "input noise" to "example-specific features" to avoid any confusion between its usage in our setup and the traditional notion of input noise.

---

> ### Author Response · Authors · 2024-11-23
>
> ## Regarding the rightmost plot in Figure 3:
>
> Sorry for the confusion! Figure 3 and the experiments in Section 5 are based on standard ERM in a linear regression task, and MAT is not being used here. To clarify, the main goal of Figure 3 and Section 5 is to discuss the broader implications of memorization and to highlight the need to study memorization under various setups. We aimed to provide a thought-provoking and balanced discussion on the other side of memorization, arguing that while it can sometimes harm generalization, there are situations where it plays a positive role, and ERM might even outperform methods like MAT in such cases. We will make this more clear in the next revision.
>
> ## Regarding Influence scores distributions:
>
> This is, in fact, related to the characteristics of the Waterbirds dataset. Specifically, "LB on Land" and "WB on Water" are the majority groups within their respective classes, but the dataset is also *class-imbalanced*. For example, the "LB on Land" group has 3,498 samples, while "WB on Water" has only 1,057 samples.
>
> As expected, memorization is more severe for smaller groups because there are fewer samples to support feature learning—i.e., the model focuses more on learning features that represent larger groups. This makes memorization less severe for "LB on Land" compared to "WB on Water." This is why, in Figure 2, the self-influence score is higher for "WB on Water" than for "LB on Land." It also supports the idea that memorization is inversely related to group size: smaller groups are more vulnerable to memorization effects.
>
> We have also extended our self-influence analysis for GroupDRO in Appendix G (highlighted in green for better visibility) titled "Additional Influence-Score Experiments" to provide a direct comparison. The results confirm that MAT is more effective at reducing self-influence scores across all subpopulations (groups). GroupDRO also demonstrates some reduction in the self-influence scores, particularly for minority groups. This further supports the idea that memorization limits generalization and that mitigating it contributes to improved generalization in subpopulation shift settings.
>
> ---
> Thanks again for your review and feedback. We hope our responses help address your concerns and show the value of our work. Let us know if you have any other questions, we’d be happy to discuss further!

---

> ### Author Response · Authors · 2024-11-25
> **Follow-Up on Rebuttal Review and Clarifications**
>
> Dear Reviewer,
>
> Thank you once again for taking the time to provide your valuable feedback on our paper. With the discussion phase nearing its end, and to ensure that our responses are fully considered, we request that you kindly confirm whether you’ve had a chance to review them.
>
> We hope our responses have addressed your concerns and clarified any ambiguities. If there are any further questions or suggestions, we would be glad to address them promptly.
>
> Additionally, if our responses have adequately resolved your concerns, we kindly ask you to consider reflecting this in your review score.
>
> Thank you!

---

> > ### Comment · Reviewer_USfi · 2024-11-27
> >
> > I appreciate your detailed responses! They have improved my understanding of your work and the context in which MAT operates.
> >
> > The performance of MAT on benchmark datasets is promising. However, I believe that your paper would be substantially strengthened by providing experiments where MAT is applied to real-world datasets. In these real-world datasets, there may be no obvious or clear segregation of groups and the inherent noise may be more complex (e.g., it may obstruct or alter features that are relevant for prediction). I believe that these experiments would further support MAT's utility.

---

> > > ### Author Response · Authors · 2024-12-01
> > >
> > > Thanks for reading our response! Happy it clarified how MAT operates. We'd like to highlight three points:
> > >
> > > - Aside from Waterbirds, other datasets, CelebA, MultiNLI, and Civil Comments are real-world problems. These are standard benchmakrs in the literature, and as you highlighted: "the performance of MAT on benchmark datasets is promising".
> > >
> > > - Per your suggestions, we added new sections in Appendices F, G, and H. These include an analysis of self-influence scores (Appendix G), results on baselines (Appendix F), and a new multi-spurious feature setup in Appendix H where MAT achieves near-perfect test accuracy across all groups (Figure 7).
> > >
> > > - Fianlly, beyond MAT itself, we believe the paper's first contribution—understanding the nuanced relationship between memorization and generalization—**is equally significant and hope it is properly weighted in your evaluation as well**.
> > >
> > > We appreciate your time again and hope you’ll reconsider your score if you think our contributions merit it. Thank you!

---

> > > > ### Author Response · Authors · 2024-12-03
> > > > **Final Follow-Up**
> > > >
> > > > Dear reviewer, could you please let us know if you’ve had a chance to review our recent response? We put a lot of effort into addressing your concerns. Thanks!

---

### Official Review · Reviewer_23ej · 2024-11-04

**Soundness:** 3
**Presentation:** 2
**Contribution:** 2
**Rating:** 6
**Confidence:** 3

**Summary:**

This paper investigates the effect of the combination of spurious correlation and memorization and points out that the latter exacerbates the poor generalization caused by the former. The claim has basis on an experimental and theoretical analysis with a linear model. Furthermore, a method to mitigate the problem is proposed and its effectiveness is checked in experimental setups of subpopulation shift.

**Strengths:**

- A theoretical analysis gives us an example model where the training with noiseless data leads to a test performance asymptotic to perfect accuracy and the training with noise data leads a test performance asymptotic to perfect fitting to spurious correlation(Theorem 2.2).

- A new training loss for mitigating the harmful effect of spurious correlation without requiring annotations on spurious features is proposed based on the theoretical analysis mentioned above (Section 3.1).

- The effect of the proposed method is checked experimentally with the subpopulation shift problem. Although superiority against existing methods for this problem is not observed clearly, the improvement from naive empirical risk minimization is obvious(Table 1). Analysis with influence function is also conducted, and it is observed that memorization is suppressed by the proposed method(Figure 2).

**Weaknesses:**

- The theoretical analysis is limited to the case of a linear model.

- It is not clear to me how the logit shift proposed in 3.1 makes the training more "memorization-aware". In my understanding, the shift down-weights the gradient descent updates coming from the training samples with spurious correlation, and it should work, but it is not clear how the shift values change depending on the extent of memorization. It is preferable if a comment about this is added in 3.1.

- (L691, L694) Duplication of a reference.

- The code is not available with the first submission.

**Questions:**

- Please consider providing clarification regarding the weaknesses.
- Is it possible to extend MAT for more general spurious correlation? In general situations, it can happen that multiple spurious features may correlate with each other, and the indirect path mentioned in Section 3 may change to the one involving intermediate correrations.

---

> ### Author Response · Authors · 2024-11-24
>
> Thank you for your thoughtful and constructive review of our submission. Below, we address each of your comments and clarify aspects of the paper as requested.
>
> ## On the theoretical analysis:
> Our theoretical analysis is indeed limited to the case of a linear model, which is a common and well-established practice in the field. The primary goal of this approach is to simplify the analysis and gain interpretable insights that can then be tested in more complex scenarios. While extending the theory to non-linear models is challenging due to their complexity, many of the findings from our linear analysis carry over to real-world, non-linear setups, as demonstrated by our experimental results.
>
> ## On how logit shift enables memorization-aware training:
> As you stated, “the shift down-weights the gradient descent updates coming from the training samples with spurious correlations.” In subpopulation shift setups, these samples are typically from the majority group (e.g., *waterbird on a water background* (majority) vs. *landbird on a land background* (minority)), and this learning dynamic often leads to the memorization of minority groups. However, under the assumption that the core features are present in both groups, the goal is to promote learning these features while avoiding harmful memorization.
>
> The logit shift makes training "memorization-aware" by adjusting the model's focus based on its performance on held-out data. When a sample is heavily memorized (e.g., when the model fits noise or spurious correlations), its held-out prediction deviates significantly, and, as a result, the logit shift places greater emphasis on these samples. On the other hand, for samples where the model relies on core features, the held-out prediction aligns closely with the target, and the emphasis is lower. Thus, the extent of the logit shift implicitly depends on the degree of memorization for each sample (e.g., emphasizing minority samples), steering the model’s training toward learning invariant (i.e., core) features.
>
> We will add an explanation of this mechanism to Section 3.1 in the revised version of the paper to improve clarity. Thank you for pointing this out.
>
> ## Duplication of a reference:
> Thank you for the precise observation; we have fixed this in the current revision.
>
> ## Code:
> We will release our codebase as soon as possible in our next revision of the paper.
>
> ## On the general spurious correlation - (Multiple spurious features setup)
>
> Thank you for your interesting suggestion to explore multiple spurious features. To investigate this, we have designed an experiment and added a new section in **Appendix H** (highlighted in purple for better visibility), titled "Multiple spurious features setup". This new design is adapted from our original setup described in Section 2.1 of the paper. We introduce a second spurious feature, resulting in one main feature ($x_y$) and two spurious features ($x_{a_1}$ and $x_{a_2}$), making the task more challenging. For more details, please refer to Appendix H.
>
> We illustrate the training dynamics of ERM (with and without memorization capacity) and MAT with memorization capacity. As shown in Figure 7, MAT proves highly effective in this scenario, achieving near-perfect test accuracy across all groups. These results demonstrate the robustness of MAT in multiple spurious features setup, and we hope this addresses your questions.
>
> ---
> Thank you once again for your review and feedback. We hope our responses have addressed your concerns and highlighted the value of our work. Please feel free to reach out if you have any additional questions—we’d be glad to discuss them further!

---

> > ### Author Response · Authors · 2024-11-28
> > **Follow-Up on Rebuttal Review and Clarifications**
> >
> > Dear Reviewer,
> >
> > Thank you once again for taking the time to provide your valuable feedback on our paper. To ensure that our responses are fully considered, we kindly request you to confirm whether you have had a chance to review them.
> >
> > We hope our responses have addressed your concerns and clarified any ambiguities. If there are any further questions or suggestions, we would be glad to address them promptly.
> >
> > Additionally, if our responses have adequately resolved your concerns, we kindly ask you to consider reflecting this in your review score.
> >
> > Thank you!

---

> > > ### Author Response · Authors · 2024-12-02
> > > **Any Final Thoughts?**
> > >
> > > Dear reviewer, could you please let us know if you’ve had a chance to review our responses? We put a lot of effort into addressing your concerns. Thanks!

---

### Official Review · Reviewer_1LhZ · 2024-11-04

**Soundness:** 1
**Presentation:** 2
**Contribution:** 1
**Rating:** 5
**Confidence:** 5

**Summary:**

This paper studies memorization in the presence of spurious features and input noise in the training data. To improve generalization the paper proposes memorization aware training (MAT). MAT computes held-out predictions, obtained using XRM (prior work), and then uses the predictions to shift model logits. The shifted logits force the model to learn generalizable features, empirically improving worst-group accuracy across common distribution shift benchmarks with spurious correlations like CelebA and Waterbirds.

**Strengths:**

- The analysis in Section 5 is interesting, particularly the distinction between the noiseless and small noise setting, where the latter actually generalizes better. Essentially, the results indicate that a small amount of independent noise in the input is essential to fit the label noise. It would have been really nice to see an expanded analysis on this claim, with some real world experiments and formal analysis.

**Weaknesses:**

- My main concern with the paper is lack of novelty in the main technical claim, i.e., memorization (afforded by overparameterization)+spurious correlations = poor generalization. There are multiple works like Shah et al. 2020, and Sagawa et al. 2020 that make the same claim with supporting evidence.
- The connection between the objective (shifting the logits) and the final goal of preventing memorization is not very clear. In my understanding, shifting the logits should have the same affect as adding more weight in the loss on minority groups.
- The theoretical analysis in Section 2.1 is very similar to Sagawa et al. 2020 (An investigation of why overparameterization exacerbates spurious correlations). In the prior work, the analysis in the same toy setup shows that memorization is exacerbated by spurious correlations, and thus the results in this work are almost directly implied by the results in Sagawa et al. 2020.
- The empirical results are not very strong, since XRM+GroupDRO is comparable to MAT.

**Questions:**

- In the "ugly" memorization setting in Section 5, it is unclear to me why in the noiseless setting, the solution learnt is different from the setting with $\sigma=10^{-4}$. The class of over-parameterized networks contain both the function learnt in the $\sigma=10^{-4}$ setting and $\sigma=0$ setting. So, why is the learning biased to the bad solution?

---

> ### Author Response · Authors · 2024-11-20
>
> Thank you for taking the time to review our work. Below, we provide clarifications and hope they address your questions. We have split our response into three parts for better readability.
>
> ## On the Novelty:
>
> As stated in lines 99-100 of our paper, our setup adapts that of Sagawa et al. (2020). However, our analysis leads to fundamentally different findings that **cannot** be derived from their results. Below, we detail the core differences:
>
> 1. **Different Focus and Findings**:
>    - **Sagawa et al.** focus on the effect of over-parameterization. They show that over-parameterization leads to a bias **against** memorization ([1]) and then conclude that: *less* memorization leads to higher worst-group test error ([2]).
>    - **Our work** centers on memorization itself. **We conclude the opposite**: *more* memorization leads to higher worst-group test error ([3, 4]).
>
>     This *apparent* contradiction arises from different setups between Sagawa’s work and ours:
>
> 2. **Different Setups**:
>    - **Sagawa et al.** assume that *the spurious feature is less noisy* than the core feature ([5]). Thus, if a model captures the core feature, it must memorize *more* due to the core feature’s higher noise.
>    - **Our work** assumes *the core feature is fully correlated with the target class*, while the spurious feature is less correlated ([6]) with it. In our setup, if learning manages to focus on the core feature, it requires no memorization of data points. This adjustment reflects more realistic settings, where spurious features, despite being easier to learn, often have weaker correlations with labels. For example, in the Waterbirds dataset, the background ("water" or "land") is less correlated with the label than the foreground ("waterbird" or "landbird").
>
>
>    Both conclusions are valid within each setup. The apparent contradiction stems from Sagawa et al.'s setup in which the core feature has more noise. As noted in Sagawa et al. (Section 9), "This assumption need not always apply...". Their focus is on over-parameterization, so this setup is suitable for their analysis but limits its generalizability.
>
> 3. **Memorization as a Starting Point**:
>    The intuition that “neural networks memorize the training data, which decreases generalization” is not new and has been discussed in earlier literature, including Shah et al. (2020), Tsipras et al. (2018), and even back in 1999 in Walczak & Cerpa (1999) ([7]). Our work formalizes this and then builds upon that, introducing MAT to leverage this insight to learn an invariant predictor.

---

> ### Author Response · Authors · 2024-11-20
>
> ## On Shifting the Logits (through MAT) vs. Reweighting (through GroupDRO):
>
> While XRM+GroupDRO (per-group reweighting) and MAT (per-example logit shifting) achieve comparable performances in table 1, they are fundamentally different:
>
> 1. **Reweighting converges to max-margin while shifting changes the loss landscape**:
>
>     Any reweighting scheme, including GroupDRO, eventually falls back to ERM if example weights are non-zero. To avoid this, as noted in the original GroupDRO paper ([8]), "stronger-than-typical l2 regularization or early stopping" with surgical precision is required. Without such measures, as shown in Figure 2 of Sagawa et al., GroupDRO’s performance reverts to ERM (see Figure 2, ERM vs DRO with Standard Regularization).
>
>     This behavior arises because, greadient decent dynamics will converge to the same max-margin solution, even with different weights per example or per group. This result follows from the first part of Theorem 3 in Soudry et al. ("The Implicit Bias of Gradient Descent on Separable Data", JMLR, 2018). To see why, note that applying group weights is equivalent to repeating data points from each group a certain number of times based on the group label, which does not affect the separability assumption of the theorem. As such, gradient descent with per-group weighting will eventually converges to the max-margin solution.
>
>     In contrast, the logit-shifting changes the fixed point of gradient descent, resulting in a different predictor. To show that, **we have now added a new section in the appendix D** (highlighted in yellow for better visibility) titled "Empirical Insights into the Effects of Reweighting and Logit-Shifting". There, we show that while reweighting changes the optimization path, it ultimately converges to the max-margin solution. Logit shifts, however, modify both the trajectory and the fixed point, making shifting more robust and alleviating the need for precise early stopping.
>
> 2. **MAT is group-agnostic**:
>
>     An advantage of MAT's logit shifting over GroupDRO is its ability to operate without relying on any notion of groups. GroupDRO requires predefined (human-labeled or XRM-inferred) groups, with the number of groups dictated by the group-inference algorithm (e.g., XRM always identifies two groups). In contrast, MAT uses soft held-out predictions for individual examples, without requring any group-specific information in the logit adjustment phase.
>
>     This flexibility allows MAT to handle an arbitrary number of groups in subpopulation shift scenarios. While standard benchmarks on which we have reported our results often involve two groups, MAT’s group-agnostic approach could offer greater adaptability in real-world problems, where explicit group definitions are often unavailable.

---

> ### Author Response · Authors · 2024-11-20
>
> ## On the Role of Noise Dimensions:
>
> We would like to first clarify that the "input noise" used in our setup differs from its traditional usage, where input noise is used as a form of regularization (e.g., Bishop, 1995 [Training with Noise is Equivalent to Tikhonov Regularization]). Instead, we use input noise to model any feature that, in expectation (infinite data), is independent (thus decorrelated) of the target labels but, due to finite samples, exhibits apparent empirical correlations. For instance, in an image classification task, irrelevant pixels might appear correlated and thus be used for memorization.
>
> In Section 5.1, we do not claim that adding noise to inputs prevents overfitting in the traditional sense. Rather, we argue that memorization *through input noise dimensions* can prevent catastrophic overfitting by redirecting spurious patterns into auxiliary features, leaving core features free to learn generalizable patterns.
>
> Now to answer your question, in the noiseless setting ($ \sigma = 0 $), the model is constrained to rely solely on a single feature (along the x-axis) to minimize the training loss to zero. Regardless of the model's over-parameterization, it must memorize every detail in the training data using this single feature. This leads to catastrophic overfitting because there are no auxiliary noise dimensions available to "absorb" small variations or non-generalizable patterns..
>
> In the *small-noise* setting ($ \sigma = 1e-4 $), however, there are noise dimensions that the model can use to fit any remaining variation that the single core feature couldn't explain. Here, memorization is not harmful; instead, it helps the model fit the last bit of unexplained variation, serving as a subtle regularizer.
>
>
> ## Planned Updates
>
> To address your remarks, we will revise the draft to:
>
> - Clarify differences between our work and Sagawa et al. (2020), emphasizing our focus on memorization and differing assumptions about core and spurious feature noise.
> - Already added appendix section D (highlighted in yellow for better visibility), "Empirical Insights into the Effects of Reweighting and Logit-Shifting", to show reweighting converges to the max-margin solution while logit shifting changes the fixed point.
> - Highlight MAT’s group-agnostic advantage in the main text, emphasizing its flexibility to handle arbitrary groups without requiring predefined groups.
> - Rename "input noise" to "example-specific features" to avoid misunderstandings. Expand Section 5.1 to explain how these example-specific features prevent catastrophic overfitting.
>
> We remain at your disposal, should you have further questions. Thank you for your time and attention.
>
> ## References:
> [1] Sagawa et al. Section 5.3: "... show how the minimum-norm inductive bias translates into a bias against memorization."
>
> [2] Sagawa et al. Page 2: "... the spurious features ... entail less memorization, and therefore suffer high worst-group test error."
>
> [3] Our work, Line 67: "... the combination of spurious correlations with memorization that leads to poor generalization."
>
> [4] Our work, Figure 2: "Models trained with ERM exhibit higher self-influence scores (higher memorization) for minority subpopulations."
>
> [5] Sagawa et al., Page 9: "... the model prefers the spurious feature because it is less noisy than the core feature."
>
> [6] Our work, Lines 124-126: $ρ^{tr} = 0.9$ makes the spurious feature correlated with y, whereas the core feature is fully correlated.
>
> [7] Walczak & Cerpa (1999), *Heuristic Principles for the Design of Artificial Neural Networks*, Page 22: "... to memorize the training data set, by providing a one-to-one mapping between input values and the corresponding output values, which decreased generalization performance."
>
> [8] Sagawa et al. (2020), Distributionally Robust Neural Networks for Group Shifts

---

> > ### Comment · Reviewer_1LhZ · 2024-11-25
> >
> > Thank you for the response!
> >
> > - I appreciate the insights into logit shifting in Appendix D, where you show that empirically logit shifting does not converge to the max-margin predictor, but re-weighting does (eventually). It would be really nice to prove this result more formally in the toy setup (Setup 2.1).
> > -  Regarding differences with the claim in Sagawa et. al. (2020), I do not think their finding is ``less memorization leads to higher worst-group test error''. In fact they state: ``overparameterized models are using spurious features and simply memorizing
> > the minority groups to get zero training error'' (but worse test-error).
> > - While I agree that MAT has the potential of being group agnostic -- in the datasets used in the paper XRM+GroupDRO does similar or better than MAT on all except MNLI. To verify this claim empirically, I would encourage the authors to consider other datasets with more sub-populations (e.g., BREEDS).
> > - Thank you for the explanation about $\sigma=1e-4$ setting. If I understood correctly, the point is that small input noise can be useful to fit unexplained noise in the label?
> >
> >
> > In light of the revisions, I will raise my score (and confidence) to 5. But overall, I still feel that the empirical evaluations are somewhat limited with small gains over baselines. At the same time, the theoretical understanding is not sufficiently novel in the context of prior works, even though Section 2.1 does provide a clean formal analysis which certainly makes the paper stronger as it stands.

---

> > > ### Author Response · Authors · 2024-11-25
> > >
> > > Thank you for raising your score. We’re glad the revisions were well-received. While we understand your concerns, we believe MAT tackles key challenges in subpopulation shift and memorization with strong potential for broader impact. We’d appreciate it if you’d consider a higher score, and we’re always happy to discuss further!
> > >
> > > ### **Regarding formal analysis on Appendix D**
> > > Thank you for your suggestion. We indeed plan to pursue future work on theoretically proving deeper insights into this change in learning dynamics by logit shifting.
> > >
> > > ### **Regarding differences with the claim in Sagawa et. al. (2020)**
> > > Thank you for your comment! We would like to respectfully clarify that your interpretation of Sagawa et al. (2020) is not fully align with their findings. Their work assumes the **spurious feature is less noisy than the core feature**. Because of that, they conclude: less memorization -> prefer to learn the spurious feature -> high worst-group error. See https://ibb.co/Pj2QPfB, where we’ve included an anonymized link highlighting relevant sections from Sagawa et al. to illustrate this fact.
> > >
> > > ### **On gains over baselines and multiple spurious features setup**
> > > While XRM+GroupDRO performs similarly to MAT on our benchmarks, MAT's group-agnostic design is a key advantage for real-world scenarios. Defining groups is often far more challenging than it appears in curated datasets like Waterbirds, and MAT addresses this without requiring explicit group labels during training.
> > >
> > > Thank you for suggesting a setup with "more sub-populations". We have experimented with a new setup in **Appendix H** (highlighted in purple for better visibility), titled "Multiple Spurious Features Setup", where we adapted our design from Section 2.1 and introduced a second spurious feature ($x_{a_2}$). This setup, therefore, includes one main feature ($x_y$) and two spurious features ($x_{a_1}$ and $x_{a_2}$), resulting in more sub-populations (eight groups in total) and making the task more challenging. As shown in Figure 7, MAT proves highly effective in this scenario, achieving near-perfect test accuracy across all groups. These results demonstrate the robustness of MAT in the multiple spurious features setup, and we hope this addresses your concern. As much as time allows, we will also work to include results on other real-world datasets (e.g. BREEDS).
> > >
> > > ### **The point is that small input noise can be useful to fit unexplained noise in the label?**
> > > Yes. Memorizing "unexplained noise in the label" through the "small amount of input noise" in this setup helps improve generalization.
> > >
> > > ---
> > > Thank you!

---

> > > > ### Comment · Reviewer_1LhZ · 2024-11-27
> > > >
> > > > > On MAT's gains over baselines
> > > >
> > > > I appreciate the authors' efforts on extending MAT to the setting with multiple groups. The empirical analysis in the toy setting with two spurious features seems promising, but to judge the efficacy of MAT over other baselines, results on real-world datasets seem quite necessary.
> > > >
> > > > > Regarding the clarification on Sagawa et al.
> > > >
> > > > My understanding is that the parts where they claim "less memorization leads to worse test-group error" (highlighted in your response as well), they are referring to the memorization of the majority group. The points in the minority group are indeed memorized. As they state in their paper, gradient descent has an "inductive bias towards learning the minimum-norm model that fits the training data" (Soudry et. al, Nagarajan et. al.). Given this they claim: "Intuitively, the minimum-norm inductive bias favors less memorization in overparameterized models". If I understand correctly, by this they mean that the solution preferred is one that does not memorize the majority group but completely memorizes the minority groups. Thus, this means that **over-parameterization leads to guaranteed memorization of the minority group, and consequently worse performance on it**. This is also reflected in their **Theorem 1** which says that when the model is over-parameterized, i.e., *$N > N_0$*, we see memorization of the minority group and worse performance on it. Thus, the claim made in Sagawa et. al. is not contradictory to the one made in this paper, and in fact consistent with it.
> > > >
> > > > Soudry, Daniel, et al. "The implicit bias of gradient descent on separable data." Journal of Machine Learning Research 19.70 (2018): 1-57.
> > > >
> > > > Nagarajan, Vaishnavh, Anders Andreassen, and Behnam Neyshabur. "Understanding the failure modes of out-of-distribution generalization." arXiv preprint arXiv:2010.15775 (2020).

---

> > > > > ### Author Response · Authors · 2024-11-27
> > > > >
> > > > > Thank you for the follow-up! It seems we’re converging on similar ideas, but the key point is that our results cannot be derived from Sagawa et al. (2020). Allow us to clarify with an example:
> > > > >
> > > > > Example A consistent with Sagawa et al. 2020:
> > > > > |                            | Core feature            | Spurious Feature        |
> > > > > | -------------------------- | ----------------------- | ----------------------- |
> > > > > | Feature noise              | 20%                     | 10%                     |
> > > > > | Final training accuracy    | 100%                    | 100%                    |
> > > > > | Percentage of memorization | 20% (all from majority) | 10% (all from minority) |
> > > > >
> > > > >
> > > > > Example B consistent with our work:
> > > > > |                            | Core feature          | Spurious Feature      |
> > > > > | -------------------------- | --------------------- | --------------------- |
> > > > > | Feature noise              | 0%                    | 10%                   |
> > > > > | Final training accuracy    | 100%                  | 100%                  |
> > > > > | Percentage of memorization | 0%                    | 10% all from minority |
> > > > >
> > > > >
> > > > > In A, the setup is that "spurious feature is less noisy". So "the spurious feature entails less memorization". As a result, preventing memorization **would not** lead to better generalization.
> > > > >
> > > > > In B, the setup is that "core feature is noiseless". So "the core feature entails less memorization". As a result, preventing memorization **would** lead to better generalization.
> > > > >
> > > > > As pointed out in our earlier response, this is just an **apparent** contradiction and both results are valid within their respective contexts.
> > > > >
> > > > > The setup in B is arguably more realistic than in A. Note that if, like in A, we assume the core feature is noisier, it becomes impossible to distinguish the core feature from the spurious feature based solely on the data.
> > > > > Regardless, the key point remains: both works build on the idea that "neural networks tend to memorize unexplained examples". However, our results and the motivation for introducing MAT cannot be derived from Sagawa et al. (2020).
> > > > >
> > > > >
> > > > > Kindly let us know if this provides more clarity.

---

> > > > > > ### Author Response · Authors · 2024-12-02
> > > > > > **Any Final Thoughts?**
> > > > > >
> > > > > > Dear reviewer, could you please let us know if you’ve had a chance to review our recent response? Thanks!

---

> > > > > > > ### Author Response · Authors · 2024-12-04
> > > > > > > **Final Follow-Up**
> > > > > > >
> > > > > > > Dear Reviewer,
> > > > > > >
> > > > > > > As today is the final day of the review process, we kindly ask if you had a chance to review our latest response. We’ve worked hard to address your concerns and would appreciate it if you could reconsider your score if our contributions merit it.
> > > > > > >
> > > > > > > We also sincerely appreciate your engagement in the discussions over the past couple of rounds.
> > > > > > >
> > > > > > > Thank you!

---

### Meta-Review · Area_Chair_MPch · 2024-12-17

**Metareview:**

The paper studies the effect of memorization on spurious correlations. Specifically, the authors argue that spurious correlations are especially problematic when the minority group examples are memorized by the model. The authors demonstrate this effect with a toy theoretical model. Inspired by this observation, the authors propose MAT, a method which applies a logit shift to the output of a classifier during training. The logit shift is done via an XRM-trained [1] model. The authors show good results on standard benchmark datasets and demonstrate that MAT reduces memorization of minority group examples using self-influence scores.

Strengths:
- The paper provides interesting insights into the connection of memorization and spurious correlations
- The proposed method is relatively simple and has a small number of hyper-parameters
  + The idea of using logit shift instead of reweighting is interesting and new; the authors argue that it leads to qualitatively different optimization behavior
- The empirical results are promising

Weaknesses:
- The proposed method relies on XRM for logit shift; GroupDRO [2] in combination with XRM group labels achieves similar performance
- The novelty of the insight about memorization is not completely clear; I do not believe it was directly stated in the same way before however
- The experiments focus on fairly small-scale datasets with isolated spurious correlations; it would be exciting to see results on more realistic datasets with multiple unknown spurious features

Recommendation: I believe the paper provides interesting new insights and a new method which shows promising results. While the experiments are limited, and related insights were presented in prior work, I believe the paper provides value to the spurious correlations community. I thus recommend to accept it as a poster.

[1] Mohammad Pezeshki, Diane Bouchacourt, Mark Ibrahim, Nicolas Ballas, Pascal Vincent, and David
Lopez-Paz. Discovering environments with xrm.

[2] Shiori Sagawa, Aditi Raghunathan, Pang Wei Koh, and Percy Liang. An investigation of why
overparameterization exacerbates spurious correlations.

**Additional Comments On Reviewer Discussion:**

The reviewers raised the following concerns:
- Novelty of the insights about memorization
- Unclear connection between insights and method
  + Unclear why MAT helps reduce memorization more than other methods
- No experiments on realistic datasets
- Limited improvements relative to baseline

The authors provided detailed responses and engaged with reviewers. They added new sections to the paper and new experiments. In particular, the authors added analysis of memorization in group DRO, optimization for logit shift vs reweighting etc. There was also a detailed conversation about the interpretation of results in Sagawa et al [2] compared to the paper under review. Generally, I believe the authors resolved all of the significant concerns raised by the reviewers.

---

### Decision · Program_Chairs · 2025-01-22

Accept (Poster)